# Cerebrospinal fluid metabolomics identifies 19 brain-related phenotype associations

Daniel J. Panyard[1], Kyeong Mo Kim[2], Burcu F. Darst [3], Yuetiva K. Deming [1,4,5], Xiaoyuan Zhong[6], Yuchang Wu[6], Hyunseung Kang[7], Cynthia M. Carlsson[4,5,8], Sterling C. Johnson[4,5,8], Sanjay Asthana[4,5,8], Corinne D. Engelman[1,9] & Qiongshi Lu [6,7,9 ✉]

The study of metabolomics and disease has enabled the discovery of new risk factors, diagnostic markers, and drug targets. For neurological and psychiatric phenotypes, the cerebrospinal fluid (CSF) is of particular importance. However, the CSF metabolome is difficult to study on a large scale due to the relative complexity of the procedure needed to collect the fluid. Here, we present a metabolome-wide association study (MWAS), which uses genetic and metabolomic data to impute metabolites into large samples with genome-wide association summary statistics. We conduct a metabolome-wide, genome-wide association analysis with 338 CSF metabolites, identifying 16 genotype-metabolite associations (metabolite quantitative trait loci, or mQTLs). We then build prediction models for all available CSF metabolites and test for associations with 27 neurological and psychiatric phenotypes, identifying 19 significant CSF metabolite-phenotype associations. Our results demonstrate the feasibility of MWAS to study omic data in scarce sample types.

[1] Department of Population Health Sciences, University of Wisconsin-Madison, 610 Walnut Street, 707 WARF Building, Madison, WI 53726, USA. [2] Department of Biotechnology, Yonsei University, 50 Yonsei-ro Seodaemun-gu, Seoul 03722, Republic of Korea. [3] Center for Genetic Epidemiology, Keck School of Medicine, University of Southern California, 1450 Biggy Street, Los Angeles, CA 90033, USA. [4] Wisconsin Alzheimer's Disease Research Center, University of Wisconsin-Madison, 600 Highland Avenue, J5/1 Mezzanine, Madison, WI 53792, USA. [5] Department of Medicine, University of Wisconsin-Madison, 1685 Highland Avenue, 5158 Medical Foundation Centennial Building, Madison, WI 53705, USA. [6] Department of Biostatistics and Medical Informatics, University of Wisconsin-Madison, WARF Room 201, 610 Walnut Street, Madison, WI 53726, USA. [7] Department of Statistics, University of Wisconsin-Madison, 1300 University Avenue, Madison, WI 53706, USA. [8] William S. Middleton Memorial Veterans Hospital, 2500 Overlook Terrace, Madison, WI 53705, USA. [9] These authors jointly supervised this work: Corinne D. Engelman, Qiongshi Lu. ✉email: qlu@biostat.wisc.edu

In recent years, the study of metabolomics has yielded novel insights into a variety of complex diseases, including diabetes[1], obesity[2], cancer[3], and Alzheimer's disease (AD)[4]. The identification of disease-associated metabolites can shed light on mechanisms contributing to disease and reveal biomarkers that can be used for diagnosis and prognosis.

To date, most metabolomics studies in humans have focused on more accessible sample types, such as blood or urine. However, for psychiatric and nervous system disorders, the cerebrospinal fluid (CSF) is of particular relevance[5,6]. CSF is in direct contact with the brain and spinal cord and is separated from the blood by the blood–brain barrier; as such, CSF may more directly reflect physiological changes occurring in the central nervous system (CNS) than other sample types. In AD, for example, CSF is the source of some of the most powerful biomarkers for disease onset and progression, including amyloid-beta (Aβ) and phosphorylated tau[7].

The difficulty in studying the CSF metabolome is that acquiring CSF samples is more challenging than blood or urine samples, requiring a lumbar puncture (LP), thus making CSF samples a rare and valuable resource, particularly those from healthy participants. This small sample size, however, makes the detection of changes in the CSF metabolome during disease progression a logistical and statistical challenge. Transcriptome-wide association studies (TWAS) have been a successful approach to dealing with such issues for gene expression[8]. Using a reference panel of genotype and gene expression measurements to model the regulatory machinery of the genetically regulated component of gene expression, TWAS allows for the estimation of potentially causal gene–disease associations in datasets where only genetic information is present, circumventing the need to collect gene expression data with every disease-focused dataset[9]. TWAS and related methods have been successfully used with a diversity of phenotypes, including autoimmune diseases[9], schizophrenia[10], and AD[11]. Building on the success of TWAS, we introduce here a metabolome-wide association study (MWAS) that combines the richness of a scarce resource study (e.g., a CSF metabolome study) with the accessibility and scale of large, publicly available genome-wide association study (GWAS) summary statistics.

The general outline of our MWAS approach is as follows: (1) identify single-nucleotide polymorphism (SNP)-metabolite associations; (2) build metabolite prediction models using genotypes; (3) test metabolite–phenotype associations with publicly available GWAS summary statistics. Step 1 is used to demonstrate that SNP-metabolite associations do indeed exist and thus justify the building of metabolite prediction models in step 2 on a cohort where both genotype and metabolite data are present. Step 3 uses the prediction models in conjunction with publicly available GWAS summary statistics on neurological and psychiatric phenotypes to test metabolite–phenotype associations. The advantage of MWAS is that it allows for this metabolite–phenotype association testing to occur in GWAS datasets where only genotypes and phenotypes (not metabolites) were originally measured.

Using MWAS, we expanded upon the limited research on the genetics of the CSF metabolome[12] by conducting a CSF metabolome-wide GWAS and then used the results to build CSF metabolite prediction models from genetic information to study the association of CSF metabolites with a variety of brain-related phenotypes using GWAS summary statistics. The CSF metabolome-wide GWAS identified 16 significant genotype–metabolite associations and was then used in MWAS to identify 19 significant CSF metabolite–brain phenotype associations, demonstrating the feasibility of MWAS for the analysis of omics data in scarce sample types.

## Results

**Cohort description and quality control.** The primary data for this study came from two different longitudinal cohort studies of AD with available CSF metabolomics and genotype data: the Wisconsin Alzheimer's Disease Research Center (WADRC) and Wisconsin Registry for Alzheimer's Prevention (WRAP) studies[13,14]. To improve the generalizability of this analysis, only data from cognitively healthy participants were used. The two study cohorts after data cleaning were similar demographically. The mean age at CSF draw was in the early to mid-60s for both cohorts (64.7 in WADRC, 62.0 in WRAP) while the sex distribution was about two-thirds female for both cohorts (63.2% in WADRC, 66.2% in WRAP) (Supplementary Data 1 and Table 1). Imputation and stringent quality control were performed on both the CSF metabolite and genotype data, resulting in a final dataset of 291 baseline visits of unrelated European-ancestry individuals with 338 CSF metabolites (Supplementary Data 1, Supplementary Tables 1–2).

**Table 1 Significant CSF metabolite–phenotype associations from BADGERS.**

| Metabolite | Phenotype | Z score | P | Q value |
|---|---|---|---|---|
| Ethylmalonate | Schizophrenia | 4.14 | 3.46E-05 | 1.22E-03 |
| Ethylmalonate | Smoking initiation | 4.22 | 2.49E-05 | 2.64E-03 |
| Cysteinylglycine | Alcoholism (drinks per week) | 3.24 | 1.20E-03 | 4.22E-02 |
| 2-hydroxy-3-methylvalerate | Schizophrenia | −3.16 | 1.58E-03 | 2.09E-02 |
| N-delta-acetylornithine | Alcoholism (drinks per week) | −4.49 | 7.05E-06 | 7.47E-04 |
| N-delta-acetylornithine | Cognitive performance | 4.83 | 1.34E-06 | 1.42E-04 |
| N-delta-acetylornithine | Schizophrenia | −5.26 | 1.42E-07 | 1.50E-05 |
| Glutaroylcarnitine (C5) | Cognitive performance | −3.72 | 2.01E-04 | 1.06E-02 |
| Cysteinylglycine disulfide | Sleep duration | −3.94 | 8.10E-05 | 8.59E-03 |
| N6-methyllysine | Schizophrenia | −3.76 | 1.73E-04 | 4.58E-03 |
| Alpha-tocopherol | Schizophrenia | 4.34 | 1.39E-05 | 7.39E-04 |
| Malate | ADHD | 3.51 | 4.49E-04 | 2.38E-02 |
| Malate | Schizophrenia | −3.21 | 1.31E-03 | 2.09E-02 |
| Glycerol | Alcoholism (drinks per week) | −3.29 | 9.99E-04 | 4.22E-02 |
| Orotate | ADHD | 3.61 | 3.05E-04 | 2.38E-02 |
| Guanosine | Schizophrenia | 3.47 | 5.24E-04 | 1.11E-02 |
| X-24295 | PTSD | −3.70 | 2.12E-04 | 2.25E-02 |
| X-24699 | Schizophrenia | −3.18 | 1.47E-03 | 2.09E-02 |
| Benzoate | Cognitive performance | 3.23 | 1.22E-03 | 4.30E-02 |

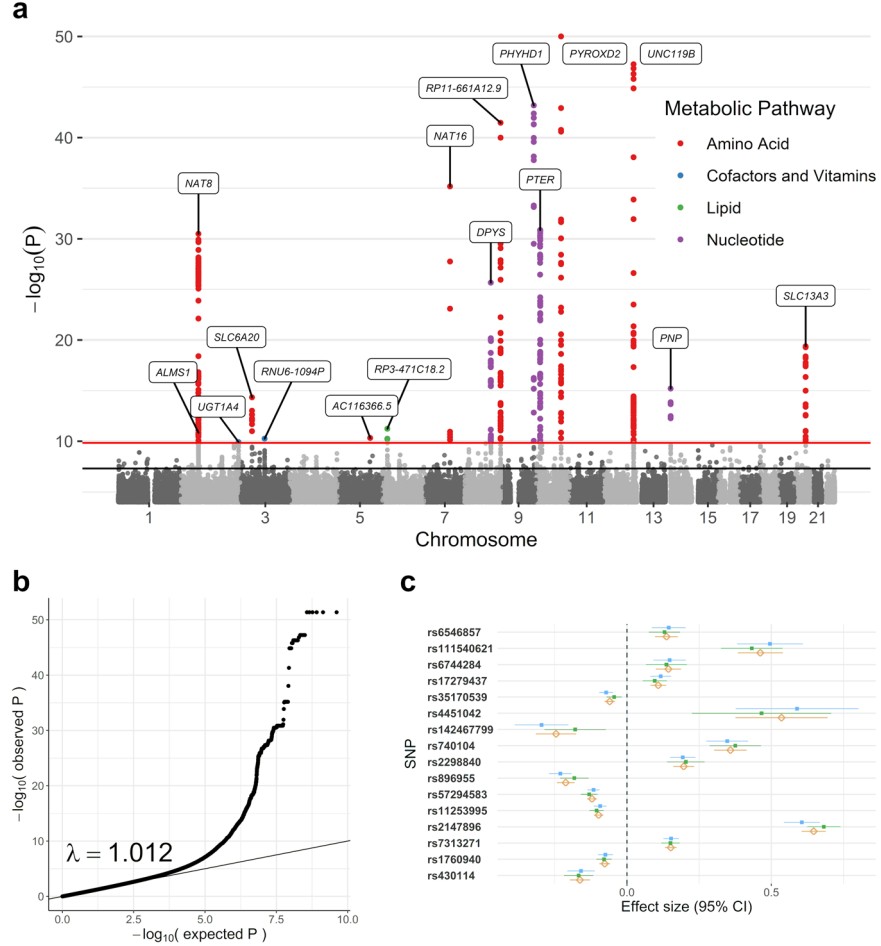

**Fig. 1 Genome-wide association study (GWAS) meta-analysis of the CSF metabolome. a** Manhattan plot of the meta-analysis across all 338 metabolites tested, with the significant SNPs colored by the metabolic pathway of the associated metabolite ($n = 291$ meta-analyzed CSF samples). Age at CSF sample, sex, genotyping batch (WADRC only), and the first five principal components were controlled for in each individual GWAS. The top SNP of each locus is labeled with the nearest gene. The horizontal lines represent the genome-wide ($5 \times 10^{-8}$, black) and Bonferroni-corrected significance thresholds ($1.48 \times 10^{-10}$, red). Data points with $P < 1 \times 10^{-50}$ for N6-methyllysine are not shown. **b** Q-Q plot based on the meta-analysis across all metabolites. **c** Forest plot of the top SNPs from each significant locus across the discovery, replication, and meta-analysis ordered by chromosome and BP position. The blue point represents the discovery GWAS, green the replication GWAS, and beige the meta-analysis. The effect size refers to the GWAS beta-effect estimate.

**GWAS of CSF metabolites.** As a first step in performing MWAS, a GWAS of CSF metabolite levels was needed (Supplementary Fig. 1). SNP-metabolite associations were estimated using GWAS conducted on both WADRC (discovery) and WRAP (replication). Both GWAS results were then meta-analyzed with METAL[15] to maximize statistical power. A total of 606 significant SNP-metabolite associations (sometimes referred to as metabolite quantitative trait loci, or mQTLs) from ten independent loci were identified in the discovery phase GWAS ($P < 1.48 \times 10^{-10}$, using the genome-wide significance threshold corrected for 338 tested metabolites), of which 488 SNPs (80.5%) and 8 loci (80%) were replicated ($P < 8.25 \times 10^{-5}$, adjusting for the 606 SNPs tested in replication). The GWAS meta-analysis identified a total of 1183 significant SNPs across 16 metabolites ($P < 1.48 \times 10^{-10}$), with one distinct genetic locus of association per metabolite (Fig. 1a; Supplementary Figs. 2–17; Supplementary Data 1, Supplementary Tables 3–4). The genomic control inflation factor across all metabolite GWAS was 1.01, indicating little evidence of inflation (Fig. 1b and Supplementary Fig. 18). The SNP effect sizes and directions were consistent across the cohorts for the top SNP at each significant locus (Fig. 1c).

Of these 16 SNP-metabolite associations, 10 (guanosine, ethylmalonate, 3-ureidopropionate, N-acetylhistidine, tryptophan betaine, N-acetyl-beta-alanine, N-delta-acetylornithine, bilirubin,

2′-O-methylcytidine, and methionine sulfone) have been previously identified in GWAS of blood, urine, or saliva samples[16–21]. Non-CSF regional association plots manually generated from publicly available summary statistics from Shin et al.[17] and Long et al.[18] were similar to corresponding CSF regional association plots, although the lead SNPs varied (Supplementary Figs. 19–25). The remaining six associations were novel, either due to the metabolite not having been reported in a GWAS previously (N-acetylglutamate, 2-hydroxyadipate, 1-ribosyl-imidazoleacetate, and N6-methyllysine) or having been analyzed previously but without identifying the same locus found here in CSF (oxalate and betaine). The top SNP, nearest gene, and brain tissue expression quantitative trait locus (eQTL) effects from each of these genotype–metabolite associations are summarized in Supplementary Data 1, Supplementary Table 3 (meta-analysis results, eQTL information across tissue types, and GWAS Catalog associations for all 1183 significant SNPs are in Supplementary Data 1, Supplementary Tables 4–6). In 9 out of the 16 loci, the eQTL effects of the top SNPs included the gene physically closest to the SNP itself.

**Metabolite prediction models.** Genome-wide prediction models were built for each CSF metabolite with independent SNPs as

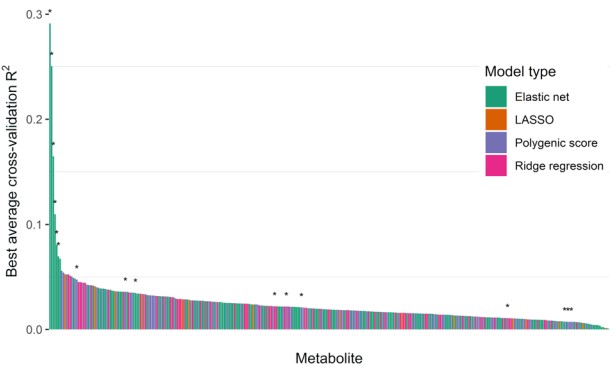

**Fig. 2 Metabolite prediction model performance.** The prediction performance of the best model for each metabolite is shown arranged in order of decreasing $R^2$. Metabolites with a significant locus from the genome-wide association study (GWAS) meta-analysis are denoted with an asterisk.

predictors (based on SNP clumping), using both models with fewer SNPs (e.g., LASSO and elastic net) and many SNPs (e.g., ridge regression and polygenic score) to allow for a diversity of possible genetic architectures, similar to gene expression prediction models used in TWAS[9]. The average predictive correlation from fourfold cross-validation was used to identify the best-performing model for each metabolite. The metabolite prediction models trained on the combined WADRC/WRAP dataset showed varying abilities to predict each metabolite (Fig. 2; Supplementary Figs. 26–27; Supplementary Data 1, Supplementary Tables 7–8). Among the top ten best-predicted metabolites from genetics, seven had a significant locus of association from the GWAS meta-analysis, and the top six were sparse models resulting from the elastic net.

Among the best-performing models for each metabolite, the $R^2$ between the predicted and actual metabolite levels ranged from 0.00083 to 0.29 (mean = 0.024, SD = 0.025), with 282 (83.4%) of the metabolites having a positive correlation and an $R^2 > 0.01$. Generally, models tended to contain 100 SNPs or more. The elastic net model was chosen as the best model type for 44.3% of the metabolites, followed by polygenic score models (28.4%), ridge regression (22.8%), and LASSO (4.5%).

**Metabolite–phenotype association testing.** The metabolite prediction models were then used to impute and test the associations of the CSF metabolites with 27 brain-related phenotypes from available GWAS summary statistics using the BADGERS approach[22]. Briefly, BADGERS functions as a summary-statistic-based TWAS-like approach that tests the association between an intermediate variable and a downstream phenotype by combining (1) a set of SNP weights for each SNP's effect on the intermediate variable with (2) GWAS summary statistics for the downstream phenotype. There were 106 models with a positive correlation and a more conservative predictive $R^2 > 0.025$. These metabolites were considered to be sufficiently well-predicted by SNPs to be included in the MWAS and were subsequently tested for association with each of 27 neurological and psychiatric phenotypes (Supplementary Data 1, Supplementary Table 9)[23–44]. We report 19 metabolite–phenotype associations that were identified at a false discovery rate (FDR) cutoff of 0.05 (Table 1; Supplementary Data 1, Supplementary Table 10; Supplementary Fig. 28). The phenotypes (and significantly associated metabolites) were schizophrenia[34] (N-delta-acetylornithine, alpha-tocopherol, ethylmalonate, N6-methyllysine, guanosine, malate, unknown metabolite X-24699, 2-hydroxy-3-methylvalerate),

cognitive performance[37] (N-delta-acetylornithine, glutaroylcarnitine [C5], benzoate), alcoholic drinks per week[40] (N-delta-acetylornithine, glycerol, cysteinylglycine), smoking behavior[40] (ethylmalonate), sleep duration[36] (cysteinylglycine disulfide), post-traumatic stress disorder (PTSD)[33] (unknown metabolite X-24295), and attention deficit hyperactivity disorder (ADHD)[26] (orotate and malate).

To demonstrate whether the MWAS associations could be seen using an alternative methodology (MR), a two-sample MR analysis was performed for the 19 significant metabolite–phenotype associations. Four effects were significant after multiple testing correction, and all four of these effects were in the same direction as predicted by BADGERS but of smaller magnitude: N-delta-acetylornithine, ethylmalonate, and N6-methyllysine with schizophrenia and N-delta-acetylornithine with cognitive performance (Supplementary Data 1, Supplementary Tables 11–12; Supplementary Fig. 29). These significant effects were all for models with only one or two SNPs used as instruments.

## Discussion

The results of this study demonstrate the feasibility of MWAS to elucidate novel metabolite–phenotype associations using metabolite prediction models built from scarce sample types and GWAS summary statistics. The first major component of MWAS was to identify SNP-metabolite associations. We identified 16 genotype–metabolite associations. As no previous metabolome-wide GWAS in the CSF had been reported to our knowledge, we assessed the validity and novelty of the results by comparing identified loci with previous GWAS of metabolites in blood, urine, and saliva[16–20,45–49]. Many of the loci discovered in this analysis of CSF metabolites replicate loci that have been previously discovered, indicating that some of the regulatory machinery of the metabolome is shared across biological compartments.

The six novel SNP-metabolite associations we identified appear to be biologically feasible based on previous research and are likely to be of general biomedical interest. The GWAS Catalog[50] reports 76 different phenotypes to be associated with these 6 loci (Supplementary Data 1, Supplementary Table 6). The chromosome 3 locus (rs17279437) associated with CSF betaine levels is closest to *SLC6A20*, a gene that has been implicated in betaine transport[51] and previously associated with N,N-dimethylglycine[45,52], which is related to betaine[53]. The SNPs associated with oxalate (ethanedioate) did not have any documented associations in the GWAS Catalog nor significant eQTLs in the Genotype-Tissue Expression project (GTEx)[54]. However, oxalate is a metabolite of the chemotherapeutic drug oxaliplatin[55], and the locus identified here is upstream of *EPHA6*, a gene that has been implicated in neuropathy from another chemotherapeutic drug, paclitaxel[56]. The locus associated with 1-ribosyl-imidazoleacetate included brain eQTLs for several genes, including *NAPRT*, which encodes an enzyme (nicotinate phosphoribosyltransferase) involved in transferring ribosyl groups[57]. The locus for N6-methyllysine includes brain eQTLs for the *PYROXD2* gene, which has been associated with other metabolites (trimethylamine and dimethylamine) in previous studies[52,58,59]. The locus associated with N-acetylglutamate included a brain eQTL for the *SLC13A3* gene that encodes sodium-dependent dicarboxylate cotransporter 3, which has been implicated as a transporter for N-acetylglutamate and for N-carbamoylglutamate, a drug used to treat N-acetylglutamate synthase deficiency[60]. The biology behind the locus for 2-hydroxyadipate was less immediately clear, but the locus includes eQTLs for the lincRNA gene *RP4-625H18.2*.

These GWAS results underscore the importance of studying scarce sample types like the CSF as they included a number of previously unreported genotype–metabolite associations. Two such metabolites (oxalate and betaine) have been previously studied in blood and urine samples, but different genetic loci were identified[16–18,21,45,48,52]. For oxalate, the strongest SNP association from blood was at $P = 1.54 \times 10^{-8}$ (rs368292858, chromosome 12, base pair (BP) 109,713,327)[18], while the strongest SNP association in CSF was stronger at $P = 5.64 \times 10^{-11}$ (rs35170539, chromosome 3, BP 96,314,015), despite having a smaller sample size. For betaine, the strongest SNP association was reported in blood at $P = 1.49 \times 10^{-19}$ (rs16876394, chromosome 5, BP 78,346,769)[17], while in the CSF the top association was at $P = 4.73 \times 10^{-15}$ (rs17279437, chromosome 3, BP 45,814,094). These CSF findings potentially represent genetic loci of control that are unique to the CSF, as they have not been identified in non-CSF studies. This partial overlap between CSF and blood QTLs for metabolites echoes that seen with studies of CSF protein levels, where only a subset (33.9%) of blood cis-protein QTLs (cis-pQTLs) were also significant CSF cis-pQTLs[61,62].

The metabolite prediction models achieved comparable performance to TWAS and imaging-wide association study (IWAS) applications. Average predictive $R^2$ values from TWAS have tended to range from 0.1 (in-sample)[9,63] to 0.02–0.05 (out-of-sample)[9]. The average in-sample $R^2$ from our MWAS was lower (0.024), perhaps as a result of metabolites being less directly controlled by the genome than gene expression and the challenge of using the entire genome for prediction rather than just cis-SNPs. Nonetheless, 83.4% of the metabolites here could still be predicted at or above the $R^2$ threshold of 0.01 used by previous studies to filter out poorly predicted gene expression values or endophenotypes[9,64], supporting the feasibility of MWAS to perform comparably to TWAS and IWAS in studying disease associations.

One benefit of this study is the insight gained into the genetic architecture of scarce sample types. The CSF metabolites studied here showed a wide range of genetic architectures as seen through the model types best able to predict them. While some metabolites with significant loci from the GWAS favored sparse models, other metabolites tended to be best predicted by a polygenic model. The performance of these predictive models also hints at the relative importance of genetics for each metabolite, which is especially helpful for datasets that are too small for effective heritability estimation, as was the case here.

The MWAS analysis identified a number of plausible CSF metabolite–phenotype associations. Three of the metabolites predicted to be associated with schizophrenia—alpha-tocopherol, N-delta-acetylornithine, and N6-methyllysine—have been implicated by previous research. Alpha-tocopherol, also known as vitamin E, is an antioxidant whose level has been noted to change in the blood during acute and chronic phases of schizophrenia[65]. Levels of N-acetylornithine, an amino acid, has been shown to differ between case and control brains in mice treated with haloperidol, an antipsychotic medication used to treat schizophrenia[66]. In humans, a recent pilot study of N-acetyl compounds found N-acetylornithine levels to be slightly but statistically insignificantly decreased in humans[67], while an MR-based study of blood metabolites identified a statistically significant decrease of N-acetylornithine levels in schizophrenia[68]. Finally, L-lysine, a related compound to N6-methyllysine, has been investigated as a potential treatment for schizophrenia[69]. Replicating metabolite associations with schizophrenia from previous research was difficult due to the lack of overlap in the metabolites, whether because the metabolite was not measured or was not well enough predicted by a genetic model to be analyzed. For instance, a study of altered metabolites in postmortem brain samples found increased hippocampal levels of glycylglycine, lactic acid, and pyridoxamine[70], but of those metabolites, only lactate was present in our dataset, and its genetic prediction model did not perform well enough for the BADGERS analysis. Other studies have reported decreased phosphatidylcholine and phosphatidylethanolamine levels in schizophrenia[71–73], but only a few such compounds were analyzed here, and none were predicted to be significantly associated with schizophrenia, which could reflect a lack of power related to the genetic predictors for these metabolites. As metabolomics technology improves and a broader array of metabolites are studied, these challenges in comparing results will lessen.

Beyond schizophrenia, other putative metabolite–phenotype associations from this analysis appeared to be feasible as well: cysteinylglycine disulfide (associated with sleep duration) is a disulfide, and disulfides have been explored as a marker of stress in obstructive sleep apnea[74], and glutaroylcarnitine (associated with cognitive performance) levels are known to be altered in glutaric acidemia type 1, which can manifest in neurological problems like dystonia[75]. N-delta-acetylornithine (associated with schizophrenia) was also associated with cognition and alcoholism. Though this particular metabolite does not appear to have been reported previously in association with cognitive performance and alcoholism, these two phenotypes have long been associated with schizophrenia[76,77].

An additional analysis made possible by the metabolite–phenotype association testing is elucidating the biology of unidentifiable metabolites. The metabolite X-24295 was significantly associated with PTSD, but little information was available on it. However, by examining the nominally significant genetic locus associated with the metabolite on chromosome 10 (BP 60,794,328-61,050,339), nearby genes and phenotypes associated with those genes were identified (genes included PHYHIPL, TRAF6P1, LINC00844, and FAM13C; phenotypes included DNA methylation, sleep duration, and QT-interval duration in Trypanosoma cruzi seropositivity). Together, these findings support and potentially shed light on the biological mechanism for metabolite X-24295's association with PTSD, as PTSD has been shown to be related to traumatic brain injury[78] (which has been associated with PHYHIPL in mice[79]), altered DNA methylation[80], and sleep disturbances[81]. These genetic annotations may also aid in the identification of the metabolite itself, as has been demonstrated by other metabolome-wide GWAS analyses[18].

The identification of promising drug targets is a major goal of metabolomics, and studies in insulin pathways[82,83], obesity[84], type 2 diabetes[85], and atherosclerosis[86] have shown the feasibility of identifying metabolites that affect disease in follow-up experimentation. In a recent review, a drug development pipeline was proposed for metabolomics-identified targets, beginning with two rounds of case-control studies with 50 or more participants[87]. Multiple studies on such a large scale may be logistically difficult to arrange for many diseases, which is where MWAS can play a key role. MWAS offers a potential alternative for the initial discovery of targets that helps avoid the need for direct metabolite analysis, instead imputing metabolites using more readily available genetic information.

One limitation of this study was the small sample size available for running the GWAS and training the metabolite prediction models. Having only a few hundred samples likely precluded the identification of some genetic loci associated with CSF metabolites; as such, the resulting predictive models could potentially be improved with larger sample size. However, even at the current sample size, the majority of metabolites could be predicted from genetics at the threshold of typical TWAS applications, and those studies have been successful in identifying gene-phenotype

associations as noted earlier. One possible explanation for the success of this GWAS in spite of the smaller sample size is that molecular traits like metabolites are more biologically proximal to the DNA and thus may be more likely to be strongly affected by genetic variants compared to complex disease. Another limitation is the training sample used for building the prediction models. Only individuals of European ancestry were studied, which may limit the generalizability of these findings to individuals of non-European ancestry, and the population was older, which may limit the ability to predict metabolite associations with phenotypes like schizophrenia that may develop in younger populations. Future applications of MWAS in diverse populations will be needed to ensure that disease associations can be identified for a broader range of populations.

Finally, the metabolite–phenotype associations identified here may not necessarily be causal. Using MWAS, an approach based on TWAS, we can identify metabolite–phenotype associations, but the identification of causal metabolite–phenotype effects requires additional assumptions to be met. The results from the MR analyses, used here to assess whether the associations from a TWAS-based approach could be replicated with an MR-based methodology, did provide significant, consistent support for some of the metabolite–phenotype associations, but we do not necessarily claim causality from these secondary analyses. The assumptions of MR would need to be met before such causal claims could be made, and there was evidence here that among the more polygenic models, there could be pleiotropy present that could violate those assumptions. For instance, the predictive model used here for guanosine included 92 SNPs. When MR-Egger regression[88] was used to estimate guanosine's effect on schizophrenia, a significant pleiotropic effect was identified (MR-Egger intercept $P = 0.0029$) that likely explains the difference in effect seen between the inverse-variance-weighted and MR-Egger results (Supplementary Data 1, Supplementary Table 11). However, as a positive example, there is support for the MR assumptions for some of the simpler metabolite models. The strongest MWAS and MR results were both for the effect of N-delta-acetylornithine on schizophrenia. The two instruments used for N-delta-acetylornithine (rs10201159, rs4934469) in the MR analysis were located near the *NAT8* and *SLC16A12* genes. *NAT8* encodes N-acetyltransferase 8, which has been associated with N-acetylornithine[89], and *SLC16A12* encodes the transporter protein solute carrier family, 16 members, 12 that is documented to transport acetate, which can be converted into acetylornithine[90]. Thus, it is plausible that the SNPs used in this model are tagging genetic loci with a causal impact on N-delta-acetylornithine levels, satisfying the MR assumption for instrument validity. Furthermore, regarding the assumption of no direct effect of the instruments on the outcome, neither of these instruments seem to be associated with schizophrenia directly in the studies used here[34,91] nor in the GWAS Catalog. As more becomes known about the functional roles of these metabolites and their related genetic loci and as datasets grow larger, the ability to assess and justify the assumptions necessary for causal inference applications will improve. Nevertheless, MWAS provides a powerful tool for the initial discovery of metabolite–phenotype associations that can then be followed up experimentally.

In conclusion, we conducted a metabolome-wide GWAS of the CSF metabolome, identifying 16 genome-wide significant associations. Some of these loci appear to be unique to the CSF based on what is currently known about the blood, urine, and saliva metabolomes. Using these genetic associations, we built genome-wide prediction models for the metabolites, achieving predictions that are comparable to those currently used by TWAS applications. We leveraged these genetic associations to conduct a summary-statistic-based MWAS on a diversity of neurological

and psychiatric phenotypes, identifying 19 significant associations, some supported by previous literature, and others novel. These findings collectively provide insight into the genetic architecture of the CSF metabolome and the roles of CSF metabolites in disease, demonstrating the potential of this framework to make inroads into the omics of scarce sample types.

## Methods

**Study participants**. This study was a secondary analysis of existing metabolomics data from CSF samples analyzed in the WADRC and WRAP cohort studies. The WADRC, previously described[92], is a longitudinal cohort study of memory, aging, and AD in middle and older-aged adults who were recruited into one of six subgroups: (1) mild late-onset AD; (2) mild cognitive impairment (MCI); (3) age-matched healthy older controls (age > 65); (4) middle-aged adults with a positive parental history of AD; (5) middle-aged adults with a negative parental history of AD; and (6) middle-aged adults with indeterminate parental history of AD. The National Institute of Neurological and Communicative Disorders and Stroke and Alzheimer's Disease and Related Disorders Association (NINCDS-ADRDA)[93] and National Institute on Aging and Alzheimer's Association (NIA-AA)[94] criteria were used for clinical diagnoses. Briefly, the inclusion criteria for WADRC participants included an age ≥45, decisional capacity, and the ability to fast from food and drink for 12 h. Briefly, exclusion criteria included the history of certain medical conditions (e.g., kidney dysfunction, congestive heart failure, major neurologic disorders other than dementia, and others), lack of a study partner, and contraindication to biomarker procedures.

The WRAP study, also previously described[13], is a longitudinal cohort study of AD in middle- and older-aged adults who were cognitively healthy at baseline, enriched for persons with a parental history of AD. Briefly, inclusion criteria include being between the ages of 40 and 65, fluent in English, able to complete neuropsychological testing, and free of health conditions that might preclude study participation. Briefly, exclusion criteria included having a diagnosis or evidence of dementia at baseline.

This study was performed as part of the Generations of WRAP (GROW) study, which was approved by the University of Wisconsin Health Sciences Institutional Review Board. Participants in the WADRC and WRAP studies provided written informed consent.

**CSF samples**. A subset of participants in both the WADRC and WRAP studies had LPs conducted to collect CSF. Similar collection protocols and staff were used in both studies to collect and store the CSF samples, which have been previously described[13,95]. Briefly, fasting CSF samples were drawn from study participants in the morning through LP and then mixed, centrifuged, aliquoted, and stored at −80 °C.

Samples were kept frozen until they were shipped overnight to Metabolon, Inc. (Durham, NC), which similarly kept samples frozen at −80 °C until analysis. Metabolon used Ultrahigh Performance Liquid Chromatography-Tandem Mass Spectrometry (UPLC-MS/MS) to conduct an untargeted metabolomics analysis of the CSF samples, processing both WADRC and WRAP simultaneously on the same platform. A total of 412 metabolites were quantified, of which 354 were identified and 58 were of unknown structural identity. The relative peak intensity was quantified for each metabolite in each sample using the area under the curve. Metabolite values were divided by the median of all values for that metabolite. Quantified metabolites were annotated with metabolite identifiers, chemical properties, and pathway information.

A total of 689 participants (532 from WADRC and 168 from WRAP) with distinct CSF samples analyzed for metabolites were initially included before metabolite quality control.

**Initial metabolite processing**. Initial metabolite quality control was performed on the 689 CSF samples, including assessment of missingness and variation, imputation, and transformation. First, the missingness of each metabolite across samples was calculated. A metabolite value may be missing for several reasons: the metabolite was not present in the sample; the metabolite was present at a level below the detection limit for that metabolite; or the metabolite was present, but there was a sample or technical issue that precluded its detection by MS. Non-xenobiotic metabolites, which were expected to be present in most samples, were removed if they were missing for ≥30% of samples. Xenobiotic metabolites, which may reasonably be completely absent in samples, were removed if they were missing for ≥80% of samples. CSF samples were removed from analysis if any sample was missing measurements for ≥40% of all metabolites in the dataset. Metabolites with an interquartile range of 0 were removed because of the limited variation available for statistical analysis. At the end of these initial processing steps, 378 metabolites across 672 samples remained.

Imputation was then performed for each cohort's samples separately. Non-xenobiotics were imputed to half the minimum value within each cohort, making the assumption that missingness was due to the metabolite being present at a level below the detection limit, while xenobiotics were not imputed since they could

feasibly be absent from the CSF. Due to consistent right-skew in the data, each metabolite was $\log_{10}$ transformed.

**Initial genotype processing**. In the WADRC cohort, samples were sent to the National Alzheimer's Coordinating Center (NACC) and genotyped by the Alzheimer's Disease Genetics Consortium (ADGC) using the Illumina HumanOmniExpress-12v1_A, Infinium HumanOmniExpressExome-8 v1-2a, or Infinium Global Screening Array v1-0 (GSAMD-24v1-0_20011747_A1) BeadChip. All genetic data underwent stringent quality control prior to imputation and analysis: variants or samples with >2% missingness, variants out of Hardy–Weinberg equilibrium (HWE) ($P < 1 \times 10^{-6}$), or samples with inconsistent genetic and self-reported sex data were removed. After pre-imputation processing with the Haplotype Reference Consortium (HRC) Checking tool[96], the genotypes were uploaded to the Michigan Imputation Server[97] where they were phased using Eagle2[98] and imputed to the HRC reference panel[99]. Variants with a low-quality score ($R^2 < 0.8$) or out of HWE were removed. Quality control was carried out separately for each genotyping chip's data. After imputation, the various chip datasets were merged together.

In the WRAP cohort, DNA was extracted from whole blood samples and genotyped using the Illumina Multi-Ethnic Genotyping Array at the University of Wisconsin Biotechnology Center[14]. Briefly, samples and variants with high missingness (>5%) and samples with inconsistent genetic and self-reported sex were removed. The resulting 1198 samples from individuals of European descent with 898,220 variants were then imputed using the Michigan Imputation Server and the HRC reference panel. Variants with a low imputation quality score ($R^2 < 0.8$), with a low minor allele frequency (MAF) (<0.001), or out of HWE were removed. Variants were annotated based on the GRCh37 assembly in both the WADRC and WRAP datasets.

**Data cleaning**. For this study, the initial dataset contained those samples with both genetic and metabolomic data (440 samples from WADRC; 165 samples from WRAP). Samples were then removed for missing age at LP, a cognitive diagnosis date more than two years from the LP date, non-European ancestry (due to lack of sample size in other ancestry groups), or having withdrawn from or being ineligible for the study. In order to maximize the generalizability of the genetic–metabolite associations, only participants who were cognitively normal at the time of the CSF draw were kept. When multiple CSF draws were available from a participant, only the first qualifying sample was retained. Similarly, if participants were related (according to identity by descent in WADRC or self-reported family relationships in WRAP), only one participant was kept per related group in order to remove genetic correlation between participants.

Metabolite missingness was then reassessed among the cleaned dataset to ensure sufficient sample size for estimating SNP effects in the GWAS: any metabolite missing ≥50% across a cohort's samples was removed. To address potential population stratification, principal component analysis (PCA) was conducted within each cohort on the subset of participants to be analyzed in the GWAS. The number of principal components (PCs) controlled for in the GWAS was selected based on a visual inspection of the scree plots, which in both cases was 5 PCs. Finally, SNPs missing ≥1% across the remaining samples in a cohort were removed, leaving 7,049,691 SNPs in WADRC and 10,494,131 SNPs in WRAP. A total of 338 metabolites across 155 samples in WADRC and 136 samples in WRAP remained after these quality control procedures.

**GWAS**. A GWAS was performed for each metabolite in each cohort using PLINK[100] (version 1.90b6.3). Linear regression with an additive genetic model was used, controlling for age at CSF draw, sex, the first five PCs, and the NACC genotyping round (for WADRC only). Post-GWAS, SNPs were removed with a MAF ≤ 0.05. A Q–Q plot and Manhattan plot were generated for each GWAS using the R package qqman[101] (version 0.1.4). The genomic inflation factor was calculated for each metabolite in each cohort using the median $\chi^2$ statistic.

In the discovery phase GWAS (WADRC cohort), a genome-wide significance threshold ($5 \times 10^{-8}$) with a Bonferroni correction for the number of metabolites tested (338) resulted in a significance threshold of $1.48 \times 10^{-10}$. The significant SNPs from the discovery phase were identified and then compared to the replication phase (WRAP cohort) to assess replication at a significance threshold of 0.05 with a Bonferroni correction for the number of significant SNPs tested from the discovery phase (606), for a final significance threshold of replication of $8.25 \times 10^{-5}$.

The discovery- and replication-phase GWAS results were then meta-analyzed using the inverse-variance-weighted approach implemented in METAL[15] (2018-08-28 version, STDERR scheme). Only SNPs present in both the discovery and replication GWAS were retained, and a genome-wide significance threshold Bonferroni-corrected for the number of metabolites analyzed ($P = 1.48 \times 10^{-10}$) was used for reporting associations. Q–Q plots, Manhattan plots, and genomic inflation factors were calculated for the meta-analysis as before, and LocusZoom[102] (version 1.4) was used to generate regional genetic association plots for a 1 Mb region around the top SNP at each significant locus, using the 1000 Genomes Nov2014 EUR population for linkage disequilibrium (LD) estimation (Supplementary Figs. 2–18).

**Evaluation of significant loci**. The significant SNPs from the meta-analysis GWAS were evaluated for the feasibility of a connection with their associated metabolites. Each SNP was annotated with the nearest gene using GENCODE[103] annotations (version 19) and known eQTLs in CNS-related tissues using GTEx[54] (version 7) (Supplementary Data 1, Supplementary Table 5). Regions around the significant SNPs were also looked up in the GWAS Catalog for previously reported phenotype associations using the R package gwasrapidd[104] (version 0.99.8) (Supplementary Data 1, Supplementary Table 6). In addition, each significant SNP-metabolite association was checked against previously published GWAS of metabolites[16–20,46,47,49] in non-CSF fluids or tissues, with a focus on publications that also used Metabolon for metabolite quantification and thus were more likely to have measured the same metabolites that were measured here. To match metabolites by name across datasets when Metabolon identifiers were not available, string-matching functions from MetaboAnalystR[105] (version 1.0.2) and stringdist[106] (version 0.9.5.5) were used to match metabolite names, which were then manually reviewed for accuracy. Each SNP-metabolite association was examined in the results of each of the non-CSF studies for the presence of the metabolite and whether the SNP association was replicated. For studies with publicly available GWAS summary statistics, LocusZoom plots were created of the CSF-significant genetic regions using the publicly available non-CSF summary statistics data to allow for a side-by-side comparison of the CSF associations and the non-CSF associations for the same metabolite at the same locus (Supplementary Figs. 19–25).

**Metabolite prediction models**. Metabolite prediction models were built and selected using fourfold cross-validation (Supplementary Fig. 26). To maximize the sample size available for training metabolite prediction models, a combined WADRC and WRAP dataset was created. Only SNPs present in both datasets, present for all individuals, and with a MAF ≥ 0.05 were retained. To account for differences in SNP annotations, SNPs were harmonized across WADRC, WRAP, and the 1000 Genomes Phase 3 CEU samples that were used as an LD reference such that all SNPs were oriented to the same strand and major/minor allele annotation. SNPs that were inconsistent across the datasets or ambiguous SNPs were removed. The combined WADRC/WRAP dataset was then partitioned evenly into four portions, with a training fold comprising three portions merged together and a testing fold comprising the remaining portion.

Within each training fold of data, PLINK was used to run a GWAS of each metabolite using a linear, additive model, controlling for age at CSF collection, sex, the top five PCs (calculated in the combined dataset), and an indicator for WADRC or WRAP genotyping round. Variance inflation factors were restricted to being less than 50 (four metabolites were excluded from further analysis due to a high inflation factor in at least one fold). The resulting fold-specific GWAS files were then clumped down to independent SNPs ($r^2 < 0.1$ within a 1000 kb window using the 1000 Genomes CEU reference panel for LD estimation) with a $P$ value threshold of 0.01 using PLINK.

Metabolite prediction models were built for each metabolite within each fold of training data. Four general model types covering a range of genetic architecture assumptions were employed: LASSO[107], elastic net[108], ridge regression[109], and polygenic score models[110]. LASSO uses L1 regularization to perform variable selection in a regression model, while ridge regression uses L2 regularization and retains all variables in the regression. Elastic net lies between LASSO and ridge regression, using a weighted combination of the L1 and L2 penalties. Polygenic score models use a weighted combination of SNPs where the weight of each SNP is based on the beta coefficient of a GWAS for the model outcome. The three penalized regression models (LASSO, elastic net, and ridge regression) were implemented using the R package glmnet[111] (version 2.0–18). An $11 \times 11$ grid of parameter combinations (lambda and alpha) was created. Lambdas ranged from $1.0 \times 10^{-5}$ to 1.0 (10 raised to exponents incremented by 0.5); alphas ranged from 0.0 to 1.0 (incremented by 0.1). Models were classified based on the alpha value (1.0 = LASSO, 0.0 = ridge regression, others = elastic net). Model predictors included all clumped SNPs and the same covariates used for the fold-specific GWAS, but the regularization penalty was only applied to the SNPs. The polygenic score models were implemented using PRSice[112] (version 2.2.4). Three $P$ value thresholds were used: 0.0001, 0.001, and 0.01.

Each fold-specific metabolite prediction model was tested on the corresponding testing fold to determine the correlation and $R^2$ between the predicted and actual metabolite values (Supplementary Fig. 27). The mean predictive correlation was taken across all folds for each model, with the highest-correlated model chosen as the best predictive model for that metabolite. For each metabolite, the type of model, mean number of SNPs used, and the presence of significant meta-analysis GWAS loci were recorded (Supplementary Data 1, Supplementary Table 7).

**Metabolite–phenotype association testing**. To test the associations between imputed metabolites and the various brain-related phenotypes, two components were needed: metabolite prediction model SNP weights and brain-related phenotype GWAS summary statistics. The best prediction models per metabolite chosen by the fourfold cross-validation described above were initially considered for the association testing. Only metabolite prediction models with a positive correlation and a mean predictive $R^2 > 0.025$ were retained. The model type and parameter settings for each metabolite's best-performing model were then run on the entire

WADRC/WRAP combined sample to generate the final model weights for all SNPs included in the model (Supplementary Data 1, Supplementary Table 8).

The phenotypes for the association analysis were chosen based on the feasibility of the CSF metabolome being relevant to the phenotype and the availability of complete GWAS summary statistics for the phenotype. The only exception was the GWAS for the AD proxy phenotype, which was developed in-house on the UK Biobank dataset as a surrogate measure for AD risk based on parental diagnosis and age at diagnosis, following previous research[113,114]. The CNS phenotypes and sources of the GWAS summary statistics[23–44] are listed in Supplementary Data 1, Supplementary Table 9. All GWAS summary statistics were harmonized to the GRCh37 SNP annotations, and orientations of the WADRC/WRAP combined dataset were used for model training. For GWAS summary statistics with only odds ratios or Z scores reported, beta-effect sizes were converted or estimated from the data provided. To maximize the SNP overlap between the model training SNPs and the GWAS summary statistics, the ImpG method[115] (FIZI package, version 0.6; Python, version 3.8) was used to impute missing SNP effect sizes in the GWAS summary statistics. Only SNPs that matched between the training data and the imputed GWAS summary statistics were retained.

The BADGERS (Biobank-wide Association Discovery using GEnetic Risk Scores) software package[22] was used to test the association of each imputed CSF metabolite with each of the GWAS summary statistics phenotypes (Supplementary Data 1, Supplementary Table 10). A Q–Q plot of all BADGERS association test results was created to assess potential inflation (Supplementary Fig. 28). An FDR was calculated using the qvalue[116] (version 2.18.0) package for each GWAS phenotype at a threshold of 0.05 to report significant associations. A Bonferroni-corrected significance threshold based on the number of metabolite (106) and phenotype (27) combinations tested with BADGERS ($P = 0.05/2862 = 1.7 \times 10^{-5}$) was additionally used to report the most conservative associations. A manual search of published literature was conducted to check for the biological feasibility of the metabolite–phenotype associations estimated by BADGERS. For metabolites marked as unknown by Metabolon, the region around the top genetic loci for the metabolite from the GWAS meta-analysis was looked up in the GWAS Catalog to identify any other associations that might inform the metabolite's role or identity.

A two-sample Mendelian Randomization was performed for each of the significant metabolite–phenotype associations from BADGERS, using the meta-analysis GWAS results for the metabolites described above and a phenotype GWAS from the IEU GWAS Database (Supplementary Data 1, Supplementary Table 11; Supplementary Fig. 29). The goal of using MR was to demonstrate whether the results from MWAS could be replicated using an alternative method (two-sample MR). Two-sample MR was used instead of one-sample MR because the neurological phenotypes studied were not available for the individuals in WADRC and WRAP on whom the CSF metabolites were measured, which is a requirement of one-sample approaches. In the implementation of two-sample MR, GWAS summary stats were used for both the CSF metabolites and the neurological phenotypes to allow for easier replication of our results by other groups who can use our GWAS summary statistics for the CSF metabolites. In selecting the SNPs to use as instruments for each metabolite, the set of independent SNPs chosen by the best metabolite prediction model in the MWAS pipeline was used for each metabolite. Since only the significant results from the BADGERS analysis were analyzed by MR, all metabolites met the minimum predictive $R^2$ criteria for being predicted by their model SNPs as was used for the BADGERS analysis (see above). When possible, the same phenotype GWAS that was used in BADGERS was used for the MR analysis; otherwise, a similar phenotype from a different study was used (Supplementary Data 1, Supplementary Table 12)[26,37,91,117–119]. The MR analysis was conducted using the TwoSampleMR[117] (version 0.5.0) package, using the Wald ratio ("mr_wald_ratio"), inverse-variance-weighted ("mr_ivw"), Egger regression ("mr_egger_regression"), and weighted median ("mr_weighted_median") methods[117]. Briefly, the Wald ratio approach was used when only a single SNP was used as an instrument, as was the case with the ethylmalonate analyses. When multiple SNPs were used as instruments, the inverse-variance-weighted, Egger regression, and weighted median approaches were used. The inverse-variance-weighted approach combines all of the ratio estimates similar to an inverse-variance-weighted, random-effects meta-analysis. The Egger regression MR approach[88] uses multiple instruments as a way to assess the presence of pleiotropy and to adjust for biases arising from a specific type of pleiotropy where the instruments' effects on the outcome are independent of their effects on the exposure. The weighted median approach[120,121] is similar to Egger regression in that it helps to address pleiotropy when using multiple SNP instruments and is robust so long as no more than 50% of the instruments are pleiotropic. This method's benefit comes from using the median effect of the instrument SNPs and weights the contribution of the SNPs by the inverse variance. Multiple two-sample MR methods were used here to compare the results from MWAS to a diversity of MR implementations. A Bonferroni-corrected significance threshold for the number of MR analyses performed (42) was used for reporting significant results ($P = 0.05/42 = 1.2 \times 10^{-3}$).

An important note for the MR analyses is that we do not necessarily claim a causal association between the metabolites and phenotypes. In order for an MR association to be causal, three assumptions must be satisfied: (1) the SNP must have an effect on the metabolite; (2) the SNP cannot have a direct effect on the phenotype; and (3) the SNP cannot be associated indirectly with the phenotype through unmeasured confounders. While the use of only SNP-metabolite prediction models that satisfy a minimum predictive $R^2$ helped address the first assumption, and while Egger regression and weighted median regression helped address bias arising from pleiotropy and association to measured confounding, the assumptions could have been violated in this context as the biological mechanisms of many of these SNPs were unknown and there were many unmeasured demographic and clinical covariates that may feasibly have affected the metabolite–phenotype relationship as well as the distribution of SNPs. Thus, the MR analyses, while useful as a way of demonstrating whether the MWAS-identified associations could be replicated from the MR framework, should not be construed as necessarily representing causal claims. However, as described in "Discussion", there was evidence to support two of the MR assumptions for the strongest association estimated by MR, which was between N-delta-acetylornithine and schizophrenia.

**Statistics and reproducibility**. General data analysis was performed primarily using R[122] (versions 3.6.0 and 3.6.1), RStudio[123] (version 1.2.1335), and the Tidyverse suite of R packages[124]. The statistical significance levels for the analyses in this manuscript included adjustments for multiple testing as appropriate based on the number of SNPs, loci, metabolites, or phenotypes tested, as described above. In the initial GWAS of CSF metabolites, a discovery, replication, and meta-analysis approach was used.

**Reporting summary**. Further information on research design is available in the Nature Research Reporting Summary linked to this article.

## Data availability

The datasets generated and analyzed during the current study may be requested from the WADRC at https://www.adrc.wisc.edu/apply-resources. Full GWAS meta-analysis summary statistics may be accessed at ftp://ftp.biostat.wisc.edu/pub/lu_group/Projects/MWAS/.

## Code availability

No new software was developed for this project; existing software is available from the sources cited.

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

## Acknowledgements

We thank Stephen Ortmann for his help with manuscript preparation and Diandra Denier for providing additional information about some of the metabolites in this study. We would like to thank WRAP and WADRC participants and the Wisconsin Alzheimer's Institute (WAI) and WADRC staff for their contributions to the WRAP and WADRC studies. Without their efforts, this research would not be possible. This research is supported by National Institutes of Health (NIH) grants R01AG27161 (Wisconsin Registry for Alzheimer Prevention: Biomarkers of Preclinical AD), R01AG054047 (Genomic and Metabolomic Data Integration in a Longitudinal Cohort at Risk for Alzheimer's Disease), R21AG067092 (Identifying Metabolomic Risk Factors in Plasma and Cerebrospinal Fluid for Alzheimer's Disease), R01AG037639 (White Matter Degeneration: Biomarkers in Preclinical Alzheimer's Disease), P30AG017266 (Center for Demography of Health and Aging), and P50AG033514 and P30AG062715 (Wisconsin Alzheimer's Disease Research Center Grant), the Helen Bader Foundation, Northwestern Mutual Foundation, Extendicare Foundation, State of Wisconsin, the Clinical and Translational Science Award (CTSA) program through the NIH National Center for Advancing Translational Sciences (NCATS) grant UL1TR000427, and the University of Wisconsin-Madison Office of the Vice Chancellor for Research and Graduate Education with funding from the Wisconsin Alumni Research Foundation. This research was supported in part by the Intramural Research Program of the National Institute on Aging. Computational resources were supported by a core grant to the Center for Demography and Ecology at the University of Wisconsin-Madison (P2CHD047873). D.J.P. was supported by an NLM training grant to the Bio-Data Science Training Program (T32LM012413). B.F.D. was supported by an NLM training grant to the Computation and Informatics in Biology and Medicine Training Program (NLM 5T15LM007359). Y.K.D. was supported by a training grant from the National Institute on Aging (T32AG000213). H.K. was supported by National Science Foundation (NSF) grant DMS-1811414 (Theory and Methods for Inferring Causal Effects with Mendelian Randomization). We thank the University of Wisconsin Madison Biotechnology Center Gene Expression Center for providing Illumina Infinium genotyping services. The GTEx Project was supported by the Common Fund of the Office of the Director of the National Institutes of Health, and by

NCI, NHGRI, NHLBI, NIDA, NIMH, and NINDS. The data used for the analyses described in this paper were obtained from the GTEx Portal on 8/1/19. The content is solely the responsibility of the authors and does not necessarily represent the official views of the NIH.

## Author contributions

D.J.P., C.D.E., and Q.L. conceived and designed the study. D.J.P. conducted the analyses and wrote the paper. B.F.D. and Y.K.D. cleaned the genetic data from the WRAP and WADRC cohort studies, respectively. K.M.K. assisted with the metabolite prediction model building. X.Z. and Y.W. assisted with the implementation of BADGERS and the preparation of the AD GWAS data. H.K. assisted with the interpretation of the MR analysis. C.M.C., S.C.J., and S.A. oversaw the WADRC and WRAP cohort studies. All authors contributed to the revision of the paper.

## Competing interests

S.C.J. served as a consultant to Roche Diagnostics in 2018. The remaining authors declare no competing interests.
