## [Peer Review File · Communications Biology]

Reviewers' comments:

Reviewer #1 (Remarks to the Author):

This is an excellent study showing novel findings of SNPs regulating metabolites in the cerebrospinal fluid (CSF) of middle aged or elderly human subjects. In addition, the authors tried to use the finding to reveal the possible metabolite-phenotype association.

In the introduction, the authors should refer to previous studies concerning the GWAS on CSF metabolomics (metabolite quantitative trait loci: mQTL). Also, I feel the authors should use the word mQTL.

The authors should add brief explanation on the elastic net, polygenic score model, ridge regression, and LASSO in the results section.

Since schizophrenia develops usually in adolescence or early adulthood, the findings may be limited by the use of metabolite-GWAS data obtained from middle aged to elderly people.

In the discussion section, the authors seem to have picked up previous data supporting their findings; however, they did not mention data not supporting them. For example, the authors should overview postmortem metabolomics studies in schizophrenia patients (e.g., PMID: 27856156).

In the discussion section, the authors should refer to previous studies examining pQTL in human CSF sample (e.g., PMID: 28031287)

In the methods section, the authors should describe at least mean age and sex distributions of the study subjects.

Reviewer #2 (Remarks to the Author):

It is an interesting and a novel work. My comments are below.

1. There is no explanation and assessments of MR assumption. Due to the sheer number of genetic variants that can be easily included in the MR approach, it is likely that the IV assumption is violated. Please look at multiple approaches introduced to detect and correct for violation of the MR assumption.

Ref.

Bowden, Jack, et al. "A framework for the investigation of pleiotropy in two-sample summary data Mendelian randomization." *Statistics in medicine* 36.11 (2017): 1783-1802.

2. To assess MR assumptions no need to know the functional roles of metabolites and the related genetic loci. Again see tests for MR assumption. The one that you need to be concerned is a loci with pleiotropic effect on the phenotype and metabolite both.

3. You have found SNP \diamond Metabolite and then metabolite \diamond Phenotype. In transcriptomic, there are co-localization approaches to assess if the gene is in the path between the SNP and phenotype. You could simply one of the co-localization test to see if the metabolites are in the path from SNP to the phenotype.

Ref.

Zhu, Zhihong, et al. "Integration of summary data from GWAS and eQTL studies predicts complex trait gene targets." *Nature genetics* 48.5 (2016): 481.

Plagnol, Vincent, et al. "Statistical independence of the colocalized association signals for type 1 diabetes and RPS26 gene expression on chromosome 12q13." *Biostatistics* 10.2 (2009): 327-334.

4. There are some metabolites such as essential amino acids and diet related metabolites as well as hormone related metabolites that are not influenced by genetic factors. Could you explain about these metabolites while you use SNPs for metabolite imputation?

5. In "Online methods", section "Metabolite prediction mode", in addition to your explanation, could you provide a chart, a diagram that explains the steps? It helps readers to follow and review the steps easier.

6. For imputation of metabolites, you used the summary statistics. Did you have access to all summary data? Usually only those that pass a certain threshold level are available in publications. But, for imputation, you need all of them not only the significance.

Ref.

Lawlor, Debbie A. "Commentary: Two-sample Mendelian randomization: opportunities and challenges." *International journal of epidemiology* 45.3 (2016): 908.

7. Imputing metabolite missing values by half of the min value might be an easy and common approach but not suggested. See multiple papers for metabolomics missing value imputation.

Ref.

Using statistical techniques and replication samples for imputation of metabolite missing values, arXiv

8. For Non-xenobiotic metabolites, you consider threshold 30% for imputation. Could you explain your reason?

It is an interesting and a novel work. My comments are below.

1. There is no explanation and assessments of MR assumption. Due to the sheer number of genetic variants that can be easily included in the MR approach, it is likely that the IV assumption is violated. Please look at multiple approaches introduced to detect and correct for violation of the MR assumption.

Ref.

Bowden, Jack, et al. "A framework for the investigation of pleiotropy in two-sample summary data Mendelian randomization." *Statistics in medicine* 36.11 (2017): 1783-1802.

2. To assess MR assumptions no need to know the functional roles of metabolites and the related genetic loci. Again see tests for MR assumption. The one that you need to be concerned is a loci with pleiotropic effect on the phenotype and metabolite both.
3. You have found SNP→Metabolite and then metabolite→Phenotype. In transcriptomic, there are co-localization approaches to assess if the gene is in the path between the SNP and phenotype. You could simply one of the co-localization test to see if the metabolites are in the path from SNP to the phenotype.

Ref.

Zhu, Zhihong, et al. "Integration of summary data from GWAS and eQTL studies predicts complex trait gene targets." *Nature genetics* 48.5 (2016): 481.

Plagnol, Vincent, et al. "Statistical independence of the colocalized association signals for type 1 diabetes and RPS26 gene expression on chromosome 12q13." *Biostatistics* 10.2 (2009): 327-334.

4. There are some metabolites such as essential amino acids and diet related metabolites as well as hormone related metabolites that are not influenced by genetic factors. Could you explain about these metabolites while you use SNPs for metabolite imputation?
5. In "Online methods", section "Metabolite prediction mode", in addition to you explanation, could you provide a chart, a diagram that explains the steps? It helps readers to follow and review the steps easier.
6. For imputation of metabolites, you used the summary statistics. Did you have access to all summary data? Usually only those that pass a certain threshold levels are available in publications. But, for imputation, you need all of them not only the significance.

Ref.

Lawlor, Debbie A. "Commentary: Two-sample Mendelian randomization: opportunities and challenges." *International journal of epidemiology* 45.3 (2016): 908.

7. Imputing metabolite missing values by half of the min value might be an easy and common approach but not suggested. See multiple papers for metabolomics missing value imputation.

Ref.

Using statistical techniques and replication samples for imputation of metabolite missing values, arxiv

8. For Non- xenobiotic metabolites, you consider threshold 30% for imputation. Could you explain your reason?

Response to reviewer comments

Note: added text is in red.

Response to Reviewer #1

- 1. In the introduction, the authors should refer to previous studies concerning the GWAS on CSF metabolomics (metabolite quantitative trait loci: mQTL). Also, I feel the authors should use the word mQTL.**

To our knowledge, only one GWAS of CSF metabolites has been conducted before, but it was limited to just monoamine metabolites (Luykx et al 2014), making our work here the first metabolome-wide GWAS in the CSF. We have now included a reference to this previous paper in the introduction:

“To do so, we expanded upon the limited research on the genetics of the CSF metabolome¹ by conducting a CSF metabolome-wide GWAS and then used the results to build CSF metabolite prediction models from genetic information to study the association of CSF metabolites with a variety of brain-related phenotypes using GWAS summary statistics.” (lines 59-63).

In the original draft of this manuscript, we avoided the QTL-based abbreviation due to the conflict between metabolite QTLs (sometimes “mQTLs” or “metQTLs”) and methylation QTLs (also sometimes “mQTLs” or “metQTLs”). However, we agree that it would be worthwhile to acknowledge this terminology in the paper because it is often seen in the literature. We have added the mQTL abbreviation to the Summary and Introduction as well as included a brief parenthetical note about the alternative terminologies in the Results so that the reader may more easily connect our work to other work that uses such QTL-based language:

“We conducted a metabolome-wide, genome-wide association analysis with 338 CSF metabolites, identifying 16 genotype-metabolite associations (metabolite quantitative trait loci, or mQTLs), 6 of which were novel.” (lines 25-28)

“A total of 606 significant SNP-CSF metabolite associations (sometimes referred to as metabolite quantitative trait loci, or mQTL)...” (lines 93-94)

2. The authors should add brief explanation on the elastic net, polygenic score model, ridge regression, and LASSO in the results section.

We appreciate this suggestion, as we agree that it would be worthwhile to include more detail about them since the high-level differences between these models are related to the different kinds of genetic architecture each one captures. A brief, high-level intuition about the differences between these models has been added to the Methods Overview in the main text, and a fuller description of the model types is now present in the relevant Methods section:

“...using both models with fewer SNPs (e.g., LASSO and elastic net) and many SNPs (e.g., ridge regression and polygenic score) to allow for a diversity of possible genetic architectures (see Online Methods for details).” (lines 83-85)

“...were employed: LASSO², elastic net³, ridge regression⁴, and polygenic score models⁵.

LASSO uses L1 regularization to perform variable selection in a regression model, while ridge regression uses L2 regularization and retains all variables in the regression. Elastic net lies between LASSO and ridge regression, using a weighted combination of the L1 and L2 penalties. Polygenic score models use a weighted combination of SNPs where the weight of each SNP is

based on the beta coefficient of a GWAS for the model outcome. The 3 penalized regression models...” (lines 491-497)

3. Since schizophrenia develops usually in adolescence or early adulthood, the findings may be limited by the use of metabolite-GWAS data obtained from middle aged to elderly people.

This comment makes a good point about a potential limitation of our findings for schizophrenia based on the sample we had available for training. In fact, more generally, this limitation about the training sample versus the target population for the application is relevant to potentially other diseases or traits beyond schizophrenia.

This population-matching issue has been encountered before. The GTEx data set for eQTLs has been widely used in TWAS applications, including to schizophrenia, but the GTEx population is predominantly older (the version 8 demographics are 83.5% age 40+)⁶. Nevertheless, GTEx has been successfully applied to the study of genes associated with schizophrenia. For instance, one study used data from GTEx and the CommonMind Consortium (CMC, also an older cohort) to study the role of gene expression in schizophrenia⁷. Despite the older nature of their gene expression data, Huckins et al. were able to identify 67 significantly associated genes, the majority of which replicated previously identified genetic loci from GWAS studies. Furthermore, as they point out in their discussion section, the “[i]dentification of SCZ-associated genes primarily expressed prenatally is notable given our adult eQTL reference panels, and may reflect common eQTL architecture across development, which is known to be partial...” Even with the imputed variable (gene expression, metabolite) being imputed using

data from an older cohort, relevant associations for a disease typically associated with a younger age may still be identified.

At any rate, the broader types of population mismatch that may affect the applicability of an analysis like MWAS are worth including. Thus, the limitations section in the discussion has been expanded to also acknowledge the potential impact of this age difference:

“...Another limitation is **the training sample used for building the prediction models**. Only individuals of European ancestry were studied, which may limit the generalizability of these findings to individuals of non-European ancestry, **and the population was older, which may limit the ability to predict metabolite associations with phenotypes like schizophrenia that may develop in younger populations**. Future applications...” (lines 288-292)

- 4. In the discussion section, the authors seem to have picked up previous data supporting their findings; however, they did not mention data not supporting them. For example, the authors should overview postmortem metabolomics studies in schizophrenia patients (e.g., PMID: 27856156).**

One of the major obstacles we faced was the lack of specific overlap in metabolites studied between our data set and those of previous metabolomics analyses. In many cases, for a specific metabolite we identified as being associated with a disease, we were unable to find any metabolomics study in that disease that also measured the same specific metabolite. So positive validation of some of our findings was not always straightforward. Similarly, assessing false negatives in our data set was also not always easy, as the significant metabolites from a previous study may not have been measured in our data set or, if they were measured, they may not have happened to be predicted well enough by genetic models to enable analysis.

These limitations were certainly part of the reason why there were not more direct comparisons possible with previous research of schizophrenia. To better illustrate this general limitation of the metabolomics literature, our findings for schizophrenia are now used to highlight some of the limitations we faced in comparing our results. We have included a description of previous schizophrenia metabolomics work (Fujii et al and others) that highlight these limitations that we faced in comparing our results to previous findings.

“Replicating metabolite associations with schizophrenia from previous research was difficult due to the lack of overlap in the metabolites, whether because the metabolite was not measured or was not well enough predicted by a genetic model to be analyzed. For instance, a study of altered metabolites in post-mortem brain samples found increased hippocampal levels of glycylglycine, lactic acid, and pyridoxamine⁸, but of those metabolites, only lactate was present in our data set, and its genetic prediction model did not perform well enough for the BADGERS analysis. Other studies have reported decreased phosphatidylcholine and phosphatidylethanolamine levels in schizophrenia⁹⁻¹¹, but only a few such compounds were analyzed here, and none were predicted to be significantly associated with schizophrenia, which could reflect a lack of power related to the genetic predictors for these metabolites. As metabolomics technology improves and a broader array of metabolites are studied, these challenges in comparing results will lessen.”

(lines 238-248)

5. In the discussion section, the authors should refer to previous studies examining pQTL in human CSF sample (e.g., PMID: 28031287)

The findings from the suggested study by Sasayama et al. in 2017¹² and the corresponding blood pQTL study it references (Lourdusamy et al. 2012¹³) make for a good comparison between

metabolite QTLs and pQTLs. In both the case of metabolomics (here) and in proteomics (the requested reference), we see a partial overlap of QTLs between the CSF and blood. This partial overlap is expected given that genetic regulation is known to be at least partly context-specific. Thus, it was not surprising that some of our QTLs for CSF metabolites were new despite the metabolites being studied in blood or other fluids.

We have added a reference to this previous work and highlighted the similarity of our findings here for CSF metabolites to the findings made previously for CSF proteins.

“These CSF findings potentially represent genetic loci of control that are unique to the CSF, as they have not been identified in non-CSF studies. **This partial overlap between CSF and blood QTLs for metabolites echoes that seen with studies of CSF protein levels, where only a subset (33.9%) of blood cis-protein QTLs (cis-pQTLs) were also significant CSF cis-pQTLs^{12,13}.**” (lines 211-213)

6. In the methods section, the authors should describe at least mean age and sex distributions of the study subjects.

Details about the demographics of the two final study cohorts are available in Supplementary Table 1 (copied below for reference). A high-level summary of these details and their implications (namely, that the two populations were similar in terms of sex and age) has now been added to the Methods Overview section as well.

“**The two study cohorts after data cleaning were similar demographically. The mean age at CSF draw was in the early 60s for both cohorts (64.7 in WADRC, 62.0 in WRAP) while the sex**

distribution was similarly about two-thirds female (63.2% in WADRC, 66.2% in WRAP) (Supplementary Table 1).” (lines 74-77)

Supplementary Table 1. Study cohort description.

Cohort	N	Female (N, %)	Age at CSF draw (mean, SD)
WADRC	155	98 (63.2%)	64.7 (6.4)
WRAP	136	90 (66.2%)	62.0 (6.6)

Response to Reviewer #2

- 1. There is no explanation and assessments of MR assumption. Due to the sheer number of genetic variants that can be easily included in the MR approach, it is likely that the IV assumption is violated. Please look at multiple approaches introduced to detect and correct for violation of the MR assumption.**

Ref.

Bowden, Jack, et al. "A framework for the investigation of pleiotropy in two-sample summary data Mendelian randomization." *Statistics in medicine* 36.11 (2017): 1783-1802.

- 2. To assess MR assumptions no need to know the functional roles of metabolites and the related genetic loci. Again see tests for MR assumption. The one that you need to be concerned is a loci with pleiotropic effect on the phenotype and metabolite both.**

This comment is certainly a fair point regarding the conclusions of causal inference techniques like MR. Because the focus of this paper was on identifying associations (not necessarily causal) between CSF metabolites and brain-related phenotypes to serve as a first step

in understanding the role of CSF metabolites, many of which have not been studied in these phenotypes, we wanted to keep the discussion similarly focused on the results of the MWAS. Since our proposed MWAS methodology is somewhat novel, we included the MR analyses as a comparison to highlight that an alternative methodology offered some support for several of our identified associations. However, we did not want to focus too much on the MR results and thereby mislead the reader into thinking the MWAS analyses should be interpreted as causal claims. Thus, our discussion of the MR analyses was kept minimal.

We agree with the point that the MR assumptions are likely violated for the models with many SNPs. A specific example of the largest pleiotropic effect (observed for guanosine's effect on schizophrenia) that was identified through the use of MR-Egger (Supplementary Table 12) has been added to the discussion section. Additional information about the limitations of the application of MR (particularly for the many-SNP models) has also been added to the discussion. Finally, the fact that the significant MR results were only for 1- or 2-SNP models has also been explicitly called out in the results section.

“...acetylmethionine with cognitive performance (Supplementary Tables 11-12). **These significant effects were all for models with only 1 or 2 SNPs used as instruments.**” (lines 168-169)

“Finally, the metabolite-phenotype associations identified here may not necessarily be causal. Even with the results of the two-sample MR providing significant, consistent support for some of the metabolite-phenotype associations, **such causal inference approaches require assumptions about the presence of pleiotropy that may not hold true¹⁴. As more SNPs are included as instruments, it becomes more likely that some SNPs have introduced pleiotropic effects that would bias the causal effect estimate. Although the significant MR results here were only for 1- or 2-SNP models, evidence for pleiotropy when using many SNPs was seen in some of the**

nonsignificant MR analyses. For instance, the predictive model used here for guanosine included 92 SNPs. When MR-Egger regression¹⁵, a technique used to assess the presence of pleiotropy, was used to estimate guanosine's effect on schizophrenia, a significant pleiotropic effect was identified ($p = 0.0029$) that likely explains the difference in effect seen between the inverse-variance-weighted and MR-Egger results (Supplementary Table 11). As more becomes known about the functional roles of these metabolites and their related genetic loci and as data sets grow larger, the ability to assess and justify the assumptions necessary for causal inference applications will improve. Nevertheless, MWAS provides a powerful tool for the initial discovery of metabolite-phenotype associations that can then be followed up experimentally.”
(lines 293-309)

3. You have found SNP \diamond Metabolite and then metabolite \diamond Phenotype. In

transcriptomic, there are co-localization approaches to assess if the gene is in the path between the SNP and phenotype. You could simply one of the co-localization test to see if the metabolites are in the path from SNP to the phenotype.

Ref.

Zhu, Zhihong, et al. "Integration of summary data from GWAS and eQTL studies predicts complex trait gene targets." *Nature genetics* 48.5 (2016): 481.

Plagnol, Vincent, et al. "Statistical independence of the colocalized association signals for type 1 diabetes and RPS26 gene expression on chromosome 12q13." *Biostatistics* 10.2 (2009): 327-334.

A co-localization analysis between the metabolite and downstream brain-related phenotype GWAS would be an alternative method to address whether the two traits are likely to share an

underlying genetic variant and therefore, as the reviewer points out, whether a common pathway of association between the metabolite and phenotype is likely. A co-localization analysis thus answers a similar question to what the MWAS addresses. One other important similarity between co-localization approaches and MWAS is that both approaches allow for the use of GWAS summary statistics as input, which is necessary in the context of this project.

There are a couple of important differences between co-localization and MWAS that led to our choice of using MWAS in this paper. First, co-localization focuses on the overlap of genetic associations without incorporating the direction of the effects and ensuring that they are the same, which would be an important consideration in assessing the evidence for a common pathway between metabolites and disease.

Second, unlike MWAS, which reports a p-value, co-localization analyses provide a posterior probability that the two traits share a common causal variant. Although there are commonly used heuristics (probabilities greater than 75% or 80%) for reporting positive results, the posterior probability is more difficult to interpret, especially in terms of type I error.

Third, and perhaps most importantly, co-localization methods require there to be some specific locus to be tested. However, many of the metabolites in this study were best predicted by a polygenic model with many independent loci. Such nuance would be missed by a co-localization analysis, making it difficult to draw much of a conclusion from the analysis because the underlying genetic architecture is not well represented. This limitation of co-localization methods is less of a problem for gene expression analyses, but for metabolites, which are not mappable to a specific genomic location like a transcript or protein, this limitation is problematic.

To demonstrate this last point, we conducted a co-localization analysis for each of the top 19 metabolite-phenotype associations reported by the MWAS. The `coloc`¹⁶ R package was used to conduct an approximate Bayes factor co-localization analysis (`coloc.abf` function). The genetic locus to test for each trait pair comprised the set of SNPs within a window of 250 kb around the lead SNP of the metabolite GWAS in the combined WADRC/WRAP sample. The results of this analysis are shown in the table below:

metabolite_name	phenotype	nsnps_in		PP.H4.abf	top_hyp_meaning	model	avg_corr_ avg_r2_b		
		_coloc					n_snp_in	_model	by_param
N6-methyllysine	scz	966	0.82		both traits associated, same variant	enet	1	0.49	0.25
N-delta-acetylornithine	alcoholism	544	0.80		both traits associated, same variant	enet	2	0.28	0.08
ethylmalonate	scz	905	0.78		both traits associated, same variant	enet	1	0.32	0.11
N-delta-acetylornithine	scz	544	0.71		both traits associated, same variant	enet	2	0.28	0.08
N-delta-acetylornithine	cognitive	544	0.66		both traits associated, same variant	enet	2	0.28	0.08
2-hydroxy-3-methylvalerate	scz	1203	0.17		neither trait has association in region	ridge	5111	0.16	0.03
ethylmalonate	smoking	905	0.08		both traits associated, but different variants	enet	1	0.32	0.11
cysteinylglycine disulfide*	sleep	1147	0.01		neither trait has association in region	ridge	5019	0.11	0.04
orotate	adhd	924	0.01		only trait 1 (metabolite) has association in region	ridge	4937	0.03	0.03
cysteinylglycine	alcoholism	873	0.00		neither trait has association in region	ridge	5097	0.13	0.03
alpha-tocopherol	scz	1184	0.00		neither trait has association in region	ridge	5159	0.06	0.03
glutaryl carnitine (C5)	cognitive	965	0.00		neither trait has association in region	enet	775	0.12	0.03
glycerol	alcoholism	837	0.00		only trait 1 (metabolite) has association in region	ridge	4941	0.15	0.05
benzoate	cognitive	709	0.00		only trait 1 (metabolite) has association in region	enet	276	0.17	0.04
X - 24295	ptsd	871	0.00		neither trait has association in region	pgs	5236	0.19	0.04
X - 24699	scz	927	0.00		neither trait has association in region	enet	190	0.13	0.04
malate	adhd	794	0.00		neither trait has association in region	enet	673	0.14	0.04
malate	scz	794	0.00		neither trait has association in region	enet	673	0.14	0.04
guanosine	scz	1103	0.00		both traits associated, but different variants	pgs	99	0.14	0.04

Table columns descriptions

metabolite_name: biochemical name of the metabolite in the colocalization analysis

phenotype: name of the phenotype in the colocalization analysis

nsnps_in_coloc: number of SNPs (present in both metabolite and phenotype GWAS) used in the colocalization analysis

PP.H4.abf: posterior probability that both traits shared the same associated variant

top_hyp_meaning: a description of the hypothesis with the greatest posterior probability according to `coloc`

model: the type of model that best predicted the metabolite

n_snps_in_model: the number of SNPs used in the best-performing prediction model for the metabolite

avg_corr_by_parameter_set: the average correlation between the predicted and observed metabolite levels for the metabolite from the cross-validation model building

avg_r2_by_parameter_set: the average R^2 between the predicted and observed metabolite levels for the metabolite from the cross-validation model building

A general distinction can be seen between the metabolites that can be best-represented by one or two genetic loci (e.g., N-delta-acetylornithine) and the metabolites that needed a polygenic model (e.g., malate). Only those metabolites with a sparse genetic architecture showed much signal for a shared causal variant with their associated phenotype from the MWAS.

Regional association plots of the GWAS results for these trait-pairs highlight this problem. As an example where co-localization works well, the selected genetic region for the N-delta-acetylornithine/schizophrenia association is shown against the GWAS results for N-delta-acetylornithine and schizophrenia below (blue line represents the genome-wide significance threshold of 5×10^{-8}):

Here, the genetic locus of association for the metabolite is strong (in fact, significant after multiple testing correction in the GWAS meta-analysis), and the included genetic region for co-localization (between the two red lines) seems to capture signal of association for both traits. This locus is just one of two genetic loci (the other is on chromosome 10) from the entire genome to achieve a predictive correlation of 0.28. The posterior probability of co-localization with schizophrenia was 71%.

As an example of when co-localization does not work well, the genetic locus of association used for the co-localization of malate and schizophrenia is shown:

The individual locus for malate is much weaker (note the scale of the Y-axis compared to the figure for N-delta-acetylornithine), but such a weaker effect of the individual locus is expected for a metabolite that was best predicted by a polygenic model. This single locus is just one of 673 independent loci that were needed to best predict the level of malate in CSF. By itself,

this locus does not adequately represent the genetic signal for malate, and as might be expected the co-localization analysis fails to find evidence that there is a shared causal variant at this locus for both malate and schizophrenia.

Thus, the MWAS approach, which allowed for the use of polygenic predictors for metabolites, was preferred over co-localization. Due to the inability to capture the polygenic nature of many of the metabolites, the results of the co-localization analysis were not included in this manuscript.

4. There are some metabolites such as essential amino acids and diet related metabolites as well as hormone related metabolites that are not influenced by genetic factors. Could you explain about these metabolites while you use SNPs for metabolite imputation?

This comment brings up a good question about whether metabolites (like essential amino acids, diet-related metabolites, and hormone-related metabolites) that are not influenced by genetic factors should be imputed using genetic information. First, regarding the biology of metabolites, even if a metabolite is not produced naturally by the body (i.e., the metabolite is exogenous), genetic variants may still impact the level of that metabolite in the CSF. Variants could impact enzymes that process or utilize the metabolite, transporters that move the metabolite into or out of the CSF space, or regulate physiological states that then impact the level of the metabolite directly or indirectly¹⁷. Thus, exogeneity does not necessarily imply lack of genetic influences. The evidence of widespread genetic impact can be seen by the non-zero heritabilities of xenobiotic metabolites in plasma, including some as high as 78.1%¹⁸.

Regardless of exogeneity, however, a metabolite may be overwhelmingly driven by non-genetic influences such that a SNP-based imputation model would seem unreasonable. This point is entirely valid and was the rationale for the R^2 -based filtering for which metabolites would be included in the MWAS analysis. Only those metabolites that could be sufficiently predicted by SNP information were included in the MWAS analysis. A brief note explaining this rationale was added to the Results section:

“There were 106 models with a positive correlation and a more conservative predictive $R^2 > 0.025$. These metabolites were considered to be sufficiently well-predicted by SNPs to be included in the MWAS and were subsequently tested for association with each of 27 neurological and psychiatric phenotypes...” (lines 151-154)

5. In “Online methods”, section “Metabolite prediction mode”, in addition to your explanation, could you provide a chart, a diagram that explains the steps? It helps readers to follow and review the steps easier.

This is a great suggestion: a visualization of the model building procedure is a useful reference to pair with the written description. A new supplementary figure has been added, now named Supplementary Figure 25: Metabolite prediction model building process. An image of this figure is provided below. The names of the other supplementary figures have been updated as appropriate due to the added figure.

Supplementary Figure 25. Metabolite prediction model building process

The sequence of steps used to build and select each metabolite prediction model are summarized. The combined WADRC and WRAP sample was used to build a variety of predictive models of varying sparsity for each metabolite. Four-fold cross-validation was used to select the best model for each metabolite based on the average predictive correlation across folds.

6. For imputation of metabolites, you used the summary statistics. Did you have access to all summary data? Usually only those that pass a certain threshold levels are available in publications. But, for imputation, you need all of them not only the significance.

Ref.

Lawlor, Debbie A. "Commentary: Two-sample Mendelian randomization: opportunities and challenges." *International journal of epidemiology* 45.3 (2016): 908.

As this comment and the reference paper point out, issues arise when incomplete summary stats or filtered SNP lists are used in an analysis like MWAS or MR. In this manuscript, only complete GWAS summary statistics with the full set of summary data available were used for analysis. This important aspect is now explicitly mentioned in the Methods section.

“The phenotypes for the association analysis were chosen based on the feasibility of the CSF metabolome being relevant to the phenotype and the availability of **complete** GWAS summary statistics for the phenotype.” (lines 522-524)

7. Imputing metabolite missing values by half of the min value might be an easy and common approach but not suggested. See multiple papers for metabolomics missing value imputation.

Ref.

Using statistical techniques and replication samples for imputation of metabolite missing values, arxiv

This comment brings up a good point about the limitations of imputation methods. We have read through the suggested article¹⁹ as well as some other related papers on metabolomics imputation methods^{20,21}.

In this paper, we imputed missing metabolite values with half of the minimum value for that metabolite according to the recommendation of Metabolon, who generated our metabolomics data. The assumption here is that missing values are due to the metabolite being present but under the platform’s minimum level of detection.

As Yazdani and Yazdani point out¹⁹, among ARIC metabolite values that were missing in one data set but present in the replication data set, the missing metabolite values were not consistently on the low end, which would suggest the reason for the missing value in the first data set may not be due to the value being below the level of detection. To what extent metabolites consistently missing in all data sets were similarly evenly distributed across the metabolite's range of values or whether they were indeed present but under the limit of detection seems to still be an open question.

Alternative methods for imputation include more nuanced approaches, like k-nearest neighbor, Bayesian PCA, or multivariate imputation by chained equation, which may outperform imputation by half of the minimum value, particularly when those values are missing completely at random and not missing due to the level of the metabolite.

The appropriateness of the imputation approach we use here ultimately depends on that assumption of the mechanism of missingness, which we have now explicitly called out in the Online Methods: Initial metabolite processing section that discusses the imputation approach.

“Imputation was then performed for each cohort's samples separately. Non-xenobiotics were imputed to half the minimum value within each cohort, **making the assumption that missingness was due to the metabolite being present at a level below the detection limit**, while xenobiotics were not imputed since they could feasibly be absent from the CSF.” (lines 380-383)

We also note that for the main metabolite-phenotype associations reported by this paper, of the 15 metabolites with a significant association, the highest level of imputation from the initial data set was 13.4% (cysteinylglycine), while 10 of the 15 metabolites had no imputed values whatsoever:

	WADRC			WRAP			Total		
	Imputed values	Total values	Percent imputed	Imputed values	Total values	Percent imputed	Imputed values	Total values	Percent imputed
N-delta-acetylmethionine	17	155	10.97	11	136	8.09	28	291	9.62
Alpha-tocopherol	0	155	0	0	136	0	0	291	0
Ethylmalonate	0	155	0	0	136	0	0	291	0
Cysteinylglycine disulfide	0	155	0	0	136	0	0	291	0
N6-methyllysine	0	155	0	0	136	0	0	291	0
Glutaryl carnitine	15	155	9.68	19	136	13.97	34	291	11.68
X-24295	3	155	1.94	5	136	3.68	8	291	2.75
Orotate	0	155	0	0	136	0	0	291	0
Malate	0	155	0	0	136	0	0	291	0
Guanosine	0	155	0	1	136	0.74	1	291	0.34
Glycerol	0	155	0	0	136	0	0	291	0
Cysteinylglycine	25	155	16.13	14	136	10.29	39	291	13.4
Benzoate	0	155	0	0	136	0	0	291	0
X-24699	0	155	0	0	136	0	0	291	0
2-hydroxy-3-methylvalerate	0	155	0	0	136	0	0	291	0

8. For Non- xenobiotic metabolites, you consider threshold 30% for imputation. Could you explain your reason?

In our initial metabolomics data set, the majority of non-xenobiotic metabolites were present for nearly all samples. However, there were a few metabolites that seemed to be missing much more frequently in the data set. Such high missingness could feasibly make sense for xenobiotics, which could represent a drug metabolite that only a few people in a cohort might be expected to have at all. However, for non-xenobiotics, metabolites missing frequently were believed to potentially represent some systematic issue with the measurement process. Thus, a threshold for exclusion of these non-xenobiotic metabolites was used that was stricter than that used for xenobiotics. The threshold chosen was selected based on the distribution of missingness (below), where the bulk of metabolites appeared to be present for the bulk of samples, with a few exceptions. The threshold of 30% was chosen as a conservative threshold to remove the 25 non-xenobiotics that were missing fairly often across our data sets.

A note about the rationale for this stricter threshold has been added:

“Non-xenobiotic metabolites, which were expected to be present in most samples, were removed if they were missing for $\geq 30\%$ of samples.” (lines 373-374)

References

1. Luykx, J. J. *et al.* Genome-wide association study of monoamine metabolite levels in human cerebrospinal fluid. *Molecular Psychiatry* **19**, 228–234 (2014).
2. Tibshirani, R. Regression Shrinkage and Selection via the Lasso. *J Roy Stat Soc B Met* **58**, 267–288 (1996).
3. Zou, H. & Hastie, T. Regularization and variable selection via the elastic net. *J Roy Stat Soc B* **67**, 301–320 (2005).
4. Hoerl, A. E. & Kennard, R. W. Ridge Regression: Biased Estimation for Nonorthogonal Problems. *Technometrics* **12**, 55–67 (1970).
5. Dudbridge, F. Power and Predictive Accuracy of Polygenic Risk Scores. *PLOS Genet* **9**, e1003348 (2013).
6. Lonsdale, J. *et al.* The Genotype-Tissue Expression (GTEx) project. *Nat Genet* **45**, 580–585 (2013).
7. Huckins, L. M. *et al.* Gene expression imputation across multiple brain regions provides insights into schizophrenia risk. *Nat Genet* **51**, 659–674 (2019).
8. Fujii, T. *et al.* Metabolic profile alterations in the postmortem brains of patients with schizophrenia using capillary electrophoresis-mass spectrometry. *Schizophr. Res.* **183**, 70–74 (2017).
9. Kaddurah-Daouk, R. *et al.* Metabolomic mapping of atypical antipsychotic effects in schizophrenia. *Mol Psychiatr* **12**, 934–945 (2007).
10. McEvoy, J. *et al.* Lipidomics Reveals Early Metabolic Changes in Subjects with Schizophrenia: Effects of Atypical Antipsychotics. *PLOS ONE* **8**, e68717 (2013).

11. Wang, D. *et al.* Metabolic profiling identifies phospholipids as potential serum biomarkers for schizophrenia. *Psychiat Res* **272**, 18–29 (2019).
12. Sasayama, D. *et al.* Genome-wide quantitative trait loci mapping of the human cerebrospinal fluid proteome. *Hum Mol Genet* **26**, 44–51 (2017).
13. Lourdasamy, A. *et al.* Identification of cis-regulatory variation influencing protein abundance levels in human plasma. *Hum Mol Genet* **21**, 3719–3726 (2012).
14. Bowden, J. *et al.* A framework for the investigation of pleiotropy in two-sample summary data Mendelian randomization. *Statistics in Medicine* **36**, 1783–1802 (2017).
15. Bowden, J., Davey Smith, G. & Burgess, S. Mendelian randomization with invalid instruments: effect estimation and bias detection through Egger regression. *Int J Epidemiol* **44**, 512–525 (2015).
16. Giambartolomei, C. *et al.* Bayesian Test for Colocalisation between Pairs of Genetic Association Studies Using Summary Statistics. *PLoS Genet* **10**, (2014).
17. Suhre, K. & Gieger, C. Genetic variation in metabolic phenotypes: study designs and applications. *Nature Reviews Genetics* **13**, 759–769 (2012).
18. Darst, B. F., Kosciak, R. L., Hogan, K. J., Johnson, S. C. & Engelman, C. D. Longitudinal plasma metabolomics of aging and sex. *Aging (Albany NY)* **11**, 1262–1282 (2019).
19. Yazdani, A. & Yazdani, A. Using statistical techniques and replication samples for imputation of metabolite missing values. *arXiv:1905.04620 [q-bio]* (2019).
20. Hrydziuszko, O. & Viant, M. R. Missing values in mass spectrometry based metabolomics: an undervalued step in the data processing pipeline. *Metabolomics* **8**, 161–174 (2012).

21. Di Guida, R. *et al.* Non-targeted UHPLC-MS metabolomic data processing methods: a comparative investigation of normalisation, missing value imputation, transformation and scaling. *Metabolomics* **12**, 93 (2016).

Reviewers' comments:

Reviewer #1 (Remarks to the Author):

I think that the authors have made revisions adequately according to the comments of the reviewers and the manuscript seems to be acceptable.

Reviewer #2 (Remarks to the Author):

The manuscript has an innovative idea, I have not seen metabolite prediction/imputation previously. The main goal of the manuscript is genetic-metabolite association. But a problem that I see is supportive approaches on page 28. Many packages are applied but no clear explanation of assumptions and how to satisfy them. It is better to focus on one package and satisfy assumptions than applying many packages without explaining the assumptions.

If the authors apply MR, they need to mention MR assumptions and how they are satisfied. I believe the application of packages on page 28 as it is currently, is not appropriate for this journal. Please see the comment #5 for more details.

Major comments.

1. Lines 66-69, Method overview is not informative, please provide a flowchart in the body to explain the steps in more details. Explain why we use summary statistics from GWAS in step 3, while we are imputing metabolites in step 2. Which summary statistics we are using from GWAS and etc. There are different approaches for using summary statistics and two-sample MR. Which one you are using? For each package, you should write a brief explanation of main steps. Only applying multiple packages without talking about the details and underlying assumptions are not good signs.
2. Page 28, you talk about two-sample summary statistics and multiple packages. But, you do not talk about the approach and the underlying assumptions. How the summary statistics are integrated with the imputed metabolites.
3. The same is true for two-sample MR. You have applied multiple packages without talking about MR assumptions and how to hold them.
4. You can run two-sample MR in different ways. You should mention which one you are using. Are both samples with summary statistics or one sample with summary data the other sample with individual level data? And please explain why you are using either of them.
5. In table 11, you have b , effect size. Please write the effect of metabolite on phenotype. Then, in this table, for the low number of instruments ($n_{\text{SNP}} \leq 2$, column "bonf_sig"=TRUE), you need to provide the LD between SNPs and the z-score for the effect of SNP on metabolite. Are the metabolites predicted by these SNPs? These are the requirements for MR/instrumental variables (here the SNPs).

For the large number of instruments, you should mention in the online method section that what the package is doing. For example, what MR Egger does with more than 3000 SNPs.

If we predict metabolites with SNPs in an accurate way, we can claim the pleiotropic effect is controlled. You should remove or edit the sentence in line 300 (the sentence with the reference 85 at the end) because the reference 85 is telling us how to control for pleiotropic action while the pleiotropy is present. However, this sentence in the manuscript does not convey that.

Answering this comment adds further values to the manuscript. Please pay attention to this

comment and answer it carefully.

6. In table 11, why different MR methods, column "Method", are used for different metabolites. Was not it possible to apply one method for all metabolites? I may miss your explanation in the manuscript.

7. A flowchart for page 28 is recommended. This is an important page of the manuscript while it does not seem that the packages are applied carefully. The way that it is written it gives the impression that only multiple packages are run without paying attention to the assumptions. Minor comments.

8. Line 81, how did you do meta-analysis, did you use any package? It should be mentioned. You mention later. But, it is good to mention it here too.

9. Line 89, you have cited BADGERS approach but, you should write a few lines about it and what it does, even if you write about it later.

10. Lines 83-84, you should provide the reference for this approach. It won't be on metabolites, but it has been applied previously on gene expression. You have referred in another place, please explicitly mention it here too.

11. Line 83, you say independent SNP as predictors. What is your definition/threshold to determine independency? Is it independent statistically? How did you select SNPs for prediction of each metabolite? Did you use their effect size from line 80?

Response to reviewer comments

Note: added text is in red.

Response to Reviewer #1

- 1. I think that the authors have made revisions adequately according to the comments of the reviewers and the manuscript seems to be acceptable.**

We thank Reviewer #1 for their time in reviewing this manuscript.

Response to Reviewer #2

- 1. Lines 66-69, Method overview is not informative, please provide a flowchart in the body to explain the steps in more details. Explain why we use summary statistics from GWAS in step 3, while we are imputing metabolites in step 2. Which summary statistics we are using from GWAS and etc. There are different approaches for using summary statistics and two-sample MR. Which one you are using? For each package, you should write a brief explanation of main steps. Only applying multiple packages without talking about the details and underlying assumptions are not good signs.**

There are a couple of different concerns brought up in this comment. Regarding the first concern about the role of the Methods Overview, since we are presenting a new general workflow for analyzing the association of metabolites with phenotypes using GWAS summary statistics and since the MWAS procedure has three discrete steps that serve as a framework for

the later presentation of the manuscript's results, we included an overview of those three key steps in the main manuscript to explicitly introduce that structure to the reader. However, we intentionally keep the Methods Overview high-level and avoid too much detail in the main body since those details are already provided in the Methods section. To address the reviewer's concern about the lack of detail in this overview section, we have added additional text to explain the role of each major step, including the reason why GWAS summary statistics are used:

“Step 1 is used to demonstrate that SNP-metabolite associations do indeed exist and thus justify the building of metabolite prediction models in step 2 on a cohort where both genotype and metabolite data are present. Step 3 uses the prediction models in conjunction with publicly available GWAS summary statistics on neurological phenotypes to test metabolite-phenotype associations. The advantage of MWAS is that it allows for this metabolite-phenotype association testing to occur in GWAS data sets where only genotypes and phenotypes (not metabolites) were originally measured. Further details on these methods may be found in the Methods section.”

(lines 69-76)

We have also added a new supplementary figure as a flow chart, as suggested, that lays out the steps of the MWAS procedure as applied in this project, copied below for reference:

Supplementary Figure 1. Overview of MWAS

The flow chart highlights the main sequence of steps in the MWAS methodology as applied in this paper. After strict genotype and metabolite quality control, a GWAS of each CSF metabolite was performed, first in WADRC as a discovery cohort and then in WRAP as a replication cohort. The results were meta-analyzed together and the resulting SNP-metabolite loci reviewed for feasibility with regional association plots, gene and eQTL annotation, and comparison to previous non-CSF metabolomics GWAS. The combined WADRC/WRAP data set was then used to build metabolite prediction models from genotypes, and the resulting models were used to test for metabolite-phenotype associations with neurological and psychiatric phenotypes in a summary-statistic-based TWAS-like method (BADGERS).

Regarding which summary statistics were used in the MWAS analysis, those details are provided in the Methods section (lines 560-562) and in Supplementary Table 9. As to why summary statistics are used in MWAS (via BADGERS) and not individual-level data, there are a few explanations for this approach. First, we were interested in studying CSF metabolites in many neurological and psychiatric phenotypes where individual-level data sets may not even exist. From the outset, the very nature of the problem we were addressing was trying to develop a method for studying CSF metabolites without directly measuring them, so analyses with individual-level data were not appropriate. Second, one of the major advantages of transcriptome-wide association study (TWAS) is that it can be performed with only summary-statistic level information (e.g., S-PrediXcan¹). The reason why it works is elaborated on in the Barbeira et al paper, but essentially it comes down to the idea that the only mathematical terms needed to calculate the metabolite-phenotype associations can be derived from GWAS summary statistics of the metabolite, GWAS summary statistics of the phenotype, and an LD reference panel.

Regarding which approach was used for the summary statistic-based association testing, BADGERS is used, as is described on lines 570-572 in the Methods Overview. An important clarification is that the reason there is no discussion of MR at this point in the manuscript is that no MR of any sort was performed as part of MWAS; MR was only used in this project as a secondary analysis (lines 176-177) to demonstrate whether the associations identified by MWAS could also be replicated using two-sample MR, a more conservative method with a nice causal interpretation. Thus, no details about MR were needed during the overview of MWAS. We appreciate this comment, as we have realized that the basis of MWAS on TWAS was not sufficiently made clear in the text.

2. Page 28, you talk about two-sample summary statistics and multiple packages. But, you do not talk about the approach and the underlying assumptions. How the summary statistics are integrated with the imputed metabolites.

To clarify, there was only one metabolite-phenotype association method/package used in MWAS: BADGERS. BADGERS uses estimated SNP effect model weights (here, from the genotype-metabolite predictive models) and combines them with GWAS summary statistics for the phenotype of interest (here, the publicly available GWAS of neurological phenotypes in Supplementary Table 9). Just like with summary-statistic-based TWAS applications (e.g., S-PrediXcan¹), BADGERS essentially imputes the intermediate variable (here, CSF metabolite levels) using genotypes in order to estimate the intermediate variable-phenotype association. In addition to the BADGERS paper cited in the manuscript that includes the details on the method, we have added the following additional text to summarize how BADGERS works:

“Briefly, BADGERS functions as a summary-statistic-based TWAS-like approach that tests the association between an intermediate variable and a downstream phenotype by combining 1) a set of SNP weights for each SNP’s effect on the intermediate variable with 2) GWAS summary statistics for the downstream phenotype.” (lines 97-100)

3. The same is true for two-sample MR. You have applied multiple packages without talking about MR assumptions and how to hold them.

The reviewer brings up an important point, which is that MR analyses only provide evidence for causality when one has met the three main assumptions of MR. Our response from the first round of review (copied for reference below) lays out the reasons why an in-depth look

at the MR assumptions for causal inference were out of scope for this manuscript, where causal inference was explicitly not claimed (Discussion, line 309) and where MR was used as a secondary analysis to demonstrate replications in association between MWAS (the focus of this paper) and a different method frequently used in metabolome-wide and proteome-wide analyses (MR).

To more explicitly convey the use of MR in this manuscript to the reader, the following text has been added to the Results section and Methods section:

“To demonstrate whether the MWAS associations could be seen using an alternative methodology (MR), a two-sample MR analysis was performed for the 19 significant metabolite-phenotype associations.” (lines 176-178)

“The goal of using MR was to demonstrate whether the results from MWAS could be replicated using an alternative method (two-sample MR).” (lines 586-587)

The text added to the manuscript in the previous round of review (see below) specifically addresses how our data here may not necessarily be causal because pleiotropy may be an issue, which might be the case with some of the MR analyses.

[response from first round of review]

This comment is certainly a fair point regarding the conclusions of causal inference techniques like MR. Because the focus of this paper was on identifying associations (not necessarily causal) between CSF metabolites and brain-related phenotypes to serve as a first step in understanding the role of CSF metabolites, many of which have not been studied in these phenotypes, we wanted to keep the discussion similarly focused on the results of the MWAS.

Since our proposed MWAS methodology is somewhat novel, we included the MR analyses as a comparison to highlight that an alternative methodology offered some support for several of our identified associations. However, we did not want to focus too much on the MR results and thereby mislead the reader into thinking the MWAS analyses should be interpreted as causal claims. Thus, our discussion of the MR analyses was kept minimal.

We agree with the point that the MR assumptions are likely violated for the models with many SNPs. A specific example of the largest pleiotropic effect (observed for guanosine's effect on schizophrenia) that was identified through the use of MR-Egger (Supplementary Table 12) has been added to the discussion section. Additional information about the limitations of the application of MR (particularly for the many-SNP models) has also been added to the discussion. Finally, the fact that the significant MR results were only for 1- or 2-SNP models has also been explicitly called out in the results section.

*“...acetylnithine with cognitive performance (Supplementary Tables 11-12). **These significant effects were all for models with only 1 or 2 SNPs used as instruments.**” (lines 168-169)*

*“Finally, the metabolite-phenotype associations identified here may not necessarily be causal. Even with the results of the two-sample MR providing significant, consistent support for some of the metabolite-phenotype associations, **such causal inference approaches require assumptions about the presence of pleiotropy that may not hold true². As more SNPs are included as instruments, it becomes more likely that some SNPs have introduced pleiotropic effects that would bias the causal effect estimate. Although the significant MR results here were only for 1- or 2-SNP models, evidence for pleiotropy when using many SNPs was seen in some of the nonsignificant MR analyses. For instance, the predictive model used here for guanosine included 92 SNPs. When MR-Egger regression³, a technique used to assess the presence of pleiotropy,***

was used to estimate guanosine's effect on schizophrenia, a significant pleiotropic effect was identified (MR-Egger intercept $p = 0.0029$) that likely explains the difference in effect seen between the inverse-variance-weighted and MR-Egger results (Supplementary Table 11). As more becomes known about the functional roles of these metabolites and their related genetic loci and as data sets grow larger, the ability to assess and justify the assumptions necessary for causal inference applications will improve. Nevertheless, MWAS provides a powerful tool for the initial discovery of metabolite-phenotype associations that can then be followed up experimentally.” (lines 293-309)

Finally, as a positive example where the MR assumptions seem to be supported, additional text has been added to the Discussion focused on the strongest association identified in the MR (between N-delta-acetyloronithine and schizophrenia):

*“However, as a positive example, there is support for the MR assumptions for some of the simpler metabolite models. The strongest MWAS and MR results were both for the effect of N-delta-acetyloronithine on schizophrenia. The two instruments used for N-delta-acetyloronithine (rs10201159, rs4934469) in the MR analysis were located near the *NAT8* and *SLC16A12* genes. *NAT8* encodes N-acetyltransferase 8, which has been associated with N-acetyloronithine⁴, and *SLC16A12* encodes the transporter protein solute carrier family 16 member 12 that is documented to transport acetate, which can be converted into acetyloronithine⁵. Thus, it is plausible that the SNPs used in this model are tagging genetic loci with a causal impact on N-delta-acetyloronithine levels, satisfying the MR assumption for instrument validity. Furthermore, regarding the assumption of no direct effect of the instruments on the outcome, neither of these instruments seem to be associated with schizophrenia directly in the studies used here^{6,7} nor in the GWAS Catalog.” (lines 327-338)*

4. You can run two-sample MR in different ways. You should mention which one you are using. Are both samples with summary statistics or one sample with summary data the other sample with individual level data? And please explain why you are using either of them.

The specific methods and packages used to conduct the two-sample MR are provided in the Methods section, copied here for reference:

“A two-sample Mendelian Randomization was performed for each of the significant metabolite-phenotype associations from BADGERS, using the meta-analysis GWAS results for the metabolites described above and a phenotype GWAS from the IEU GWAS Database (Supplementary Table 11). The goal of using MR was to demonstrate whether the results from MWAS could be replicated using an alternative method (two-sample MR)...When possible, the same phenotype GWAS that was used in BADGERS was used for the MR analysis; otherwise, a similar phenotype from a different study was used (Supplementary Table 12)⁷⁻¹². The MR analysis was conducted using the TwoSampleMR¹⁰ (version 0.5.0) package, using the Wald ratio (“mr_wald_ratio”), inverse-variance-weighted (“mr_ivw”), Egger regression (“mr_egger_regression”), and weighted median (“mr_weighted_median”) methods...Multiple two-sample MR methods were used here to compare the results from MWAS to a diversity of MR implementations. A Bonferroni-corrected significance threshold for the number of MR analyses performed (42) was used for reporting significant results ($p = 0.05 / 42 = 1.2 \times 10^{-3}$).” (lines 583-617)

To conduct one-sample MR, one must have a data set with all variables of interest measured simultaneously (here, SNP, metabolite, and phenotype). Since our data sets in

WADRC and WRAP only consisted of SNP and metabolite data without the neurological phenotypes we were interested in studying, one-sample MR was not an option. GWAS summary stats were used both for the SNP-metabolite effects and the SNP-phenotype effects. The former were input using summary statistics to enable easier replication of our work by other groups who will have immediate access to our GWAS summary statistics, and the latter were input using summary statistics since those were the only data sets available for us to use for those phenotypes. Some additional text has been added to the Methods section to explicitly explain this choice of methods:

“Two-sample MR was used instead of one-sample MR because the neurological phenotypes studied were not available for the individuals in WADRC and WRAP on whom the CSF metabolites were measured, which is a requirement of one-sample approaches. In the implementation of two-sample MR, GWAS summary stats were used for both the CSF metabolites and the neurological phenotypes to allow for easier replication of our results by other groups who can use our GWAS summary statistics for the metabolites.” (lines 587-593)

5. In table 11, you have b, effect size. Please write the effect of metabolite on phenotype. Then, in this table, for the low number of instruments ($n_{\text{SNP}} \leq 2$, column “`bonf_sig`”=TRUE), you need to provide the LD between SNPs and the z-score for the effect of SNP on metabolite. Are the metabolites predicted by these SNPs? These are the requirements for MR/instrumental variables (here the SNPs).

For the large number of instruments, you should mention in the online method section that what the package is doing. For example, what MR Egger does with more than 3000 SNPs.

If we predict metabolites with SNPs in an accurate way, we can claim the pleiotropic effect is controlled. You should remove or edit the sentence in line 300 (the sentence with the reference 85 at the end) because the reference 85 is telling us how to control for pleiotropic action while the pleiotropy is present. However, this sentence in the manuscript does not convey that.

Answering this comment adds further values to the manuscript. Please pay attention to this comment and answer it carefully.

6. In table 11, why different MR methods, column “Method”, are used for different metabolites. Was not it possible to apply one method for all metabolites? I may miss your explanation in the manuscript.

7. A flowchart for page 28 is recommended. This is an important page of the manuscript while it does not seem that the packages are applied carefully. The way that it is written it gives the impression that only multiple packages are run without paying attention to the assumptions.

Regarding the first comment about the text description of the “b” column, the meaning in Supplementary Table 11 has been changed to “**Effect estimate of the metabolite on the phenotype.**” For the LD between the SNPs for those analyses where the metabolite was predicted with 2 SNPs, there was only one such metabolite model, and that was for N-delta-acetylornithine. The two SNPs here (rs10201159, rs4934469) are located on different chromosomes (2 and 10, respectively), and thus a column describing the LD between them would not be meaningful. The Z score for the SNPs’ effect on the metabolite as determined by the GWAS meta-analysis for the metabolites can be calculated from the GWAS summary

statistics using the beta effect estimate and the SE (Supplementary Table 4 and online resource of full GWAS summary statistics as described in the Data and code availability section at ftp://ftp.biostat.wisc.edu/pub/lu_group/Projects/MWAS/). As requested, these Z scores for the models with 2 or fewer SNPs have been added to Supplementary Table 11.

As to whether these instrument SNPs predicted their metabolites, since BADGERS only tested metabolites that were predicted by their SNPs with an $R^2 > 0.025$, and since only the significant associations from BADGERS were tested with MR, all of the MR-tested metabolites were necessarily predicted by their SNPs at the same minimum threshold. However, a reasonable predictive performance of penalized linear regression (here, our CSF metabolite imputation model) may not rule out horizontal pleiotropy. It remains possible that SNPs included in the metabolite imputation models could have direct effects on the tested neurological traits.

We appreciate the chance to clarify (and make more explicit to the reader) that we are not using the MR analyses to necessarily claim causality. The goal of this paper was to identify novel metabolite-phenotype associations. Multiple MR methods were used only as a way to show how the *associations* (not causal effects) from MWAS (a TWAS-based approach) could be replicated by a fundamentally different approach (MR) and some of its more popular variations. We recognize that MR overall and the various implementations of MR have different assumptions and theoretical properties. We have added additional text to the Methods section and have rewritten part of the Discussion section to clarify why MR was used as a secondary analysis and some brief notes about these different MR implementations. However, as this paper is fundamentally focused on a TWAS-based approach, we intentionally avoid providing too much detail and emphasis on what was solely a secondary analysis (the MR analyses) in order to

avoid distracting from the main TWAS-based method (MWAS) and its findings. The updated Discussion section on the study limitations regarding causality now reads...

“Finally, the metabolite-phenotype associations identified here may not necessarily be causal. Using MWAS, an approach based on TWAS, we can identify metabolite-phenotype associations, but the identification of causal metabolite-phenotype effects requires additional assumptions to be met. The results from the MR analyses, used here to assess whether the associations from a TWAS-based approach could be replicated with an MR-based methodology, did provide significant, consistent support for some of the metabolite-phenotype associations, but we do not necessarily claim causality from these secondary analyses. The assumptions of MR would need to be met before such causal claims could be made, and there was evidence here that among the more polygenic models, there could be pleiotropy present that could violate those assumptions. For instance, the predictive model used here for guanosine included 92 SNPs. When MR-Egger regression³ was used to estimate guanosine’s effect on schizophrenia, a significant pleiotropic effect was identified (MR-Egger intercept $p = 0.0029$) that likely explains the difference in effect seen between the inverse-variance-weighted and MR-Egger results (Supplementary Table 11).” (lines 309-327)

The updated Methods section on MR now reads...

“A two-sample Mendelian Randomization was performed for each of the significant metabolite-phenotype associations from BADGERS, using the meta-analysis GWAS results for the metabolites described above and a phenotype GWAS from the IEU GWAS Database (Supplementary Table 11). The goal of using MR was to demonstrate whether the results from MWAS could be replicated using an alternative method (two-sample MR). Two-sample MR was

used instead of one-sample MR because the neurological phenotypes studied were not available for the individuals in WADRC and WRAP on whom the CSF metabolites were measured, which is a requirement of one-sample approaches. In the implementation of two-sample MR, GWAS summary stats were used for both the CSF metabolites and the neurological phenotypes to allow for easier replication of our results by other groups who can use our GWAS summary statistics for the metabolites. In selecting the SNPs to use as instruments for each metabolite, the set of independent SNPs chosen by the best metabolite prediction model in the MWAS pipeline was used for each metabolite. Since only the significant results from the BADGERS analysis were analyzed by MR, all metabolites met the minimum predictive R^2 criteria for being predicted by their model SNPs as was used for the BADGERS analysis (see above). When possible, the same phenotype GWAS that was used in BADGERS was used for the MR analysis; otherwise, a similar phenotype from a different study was used (Supplementary Table 12)⁷⁻¹². The MR analysis was conducted using the TwoSampleMR¹⁰ (version 0.5.0) package, using the Wald ratio (“mr_wald_ratio”), inverse-variance-weighted (“mr_ivw”), Egger regression (“mr_egger_regression”), and weighted median (“mr_weighted_median”) methods¹⁰. Briefly, the Wald ratio approach was used when only a single SNP was used as an instrument, as was the case with the ethylmalonate analyses. When multiple SNPs were used as instruments, the inverse-variance-weighted, Egger regression, and weighted median approaches were used. The inverse-variance-weighted approach combines all of the ratio estimates similar to an inverse-variance-weighted, random effects meta-analysis. The Egger regression MR approach³ uses multiple instruments as a way to assess the presence of pleiotropy and to adjust for biases arising from a specific type of pleiotropy where the instruments’ effects on the outcome are independent of their effects on the exposure. The weighted median approach^{13,14} is similar to Egger regression

in that it helps to address pleiotropy when using multiple SNP instruments and is robust so long as no more than 50% of the instruments are pleiotropic. The method's benefit comes from using the median effect of the instrument SNPs and weights the contribution of the SNPs by the inverse variance. Multiple two-sample MR methods were used here to compare the results from MWAS to a diversity of MR implementations. A Bonferroni-corrected significance threshold for the number of MR analyses performed (42) was used for reporting significant results ($p = 0.05 / 42 = 1.2 \times 10^{-3}$).” (lines 583-617)

Regarding the addition of a separate supplementary figure to show the MR analyses, we have added a flowchart for the overall MWAS approach (Supplementary Figure 1) to help orient the reader to the general outline of the paper. Since the MR analyses were not part of the central MWAS pipeline presented by this manuscript, we have left the MR analyses out of that figure and have instead relied on a text description of the secondary analyses that were performed, including MR.

Minor comments.

8. Line 81, how did you do meta-analysis, did you use any package? It should be mentioned. You mention later. But, it is good to mention it here too.

To clarify the package used to conduct the meta-analysis in the Methods Overview, this line now reads: “Both GWAS results were then meta-analyzed with METAL¹⁵ to maximize statistical power.” (lines 88-89)

9. Line 89, you have cited BADGERS approach but, you should write a few lines about it and what it does, even if you write about it later.

Please see the added text as part of comment #2 above, which addresses this comment.

10. Lines 83-84, you should provide the reference for this approach. It won't be on metabolites, but it has been applied previously on gene expression. You have referred in another place, please explicitly mention it here too.

To highlight the similarity of our metabolite predictive models with those used in TWAS for gene expression, we have now added the following text:

“Genome-wide prediction models were built for each CSF metabolite with independent SNPs as predictors, using both models with fewer SNPs (e.g., LASSO and elastic net) and many SNPs (e.g., ridge regression and polygenic score) to allow for a diversity of possible genetic architectures (see Methods for details), **similar to gene expression prediction models used in TWAS¹⁶.**” (lines 89-93)

For reference, the Methods text describing the details of this approach are copied below:

“Metabolite prediction models were built for each metabolite within each fold of training data. Four general model types covering a range of genetic architecture assumptions were employed: LASSO¹⁷, elastic net¹⁸, ridge regression¹⁹, and polygenic score models²⁰. LASSO uses L1 regularization to perform variable selection in a regression model, while ridge regression uses L2 regularization and retains all variables in the regression. Elastic net lies between LASSO and ridge regression, using a weighted combination of the L1 and L2 penalties. Polygenic score

models use a weighted combination of SNPs where the weight of each SNP is based on the beta coefficient of a GWAS for the model outcome. The 3 penalized regression models (LASSO, elastic net, and ridge regression) were implemented using the R package glmnet²¹ (version 2.0-18). An 11x11 grid of parameter combinations (lambda and alpha) was created. Lambdas ranged from 1.0×10^{-5} to 1.0 (10 raised to exponents incremented by 0.5); alphas ranged from 0.0 to 1.0 (incremented by 0.1). Models were classified based on the alpha value (1.0 = LASSO, 0.0 = ridge regression, others = elastic net). Model predictors included all clumped SNPs and the same covariates used for the fold-specific GWAS, but the regularization penalty was only applied to the SNPs. The polygenic score models were implemented using PRSice²² (version 2.2.4). Three p-value thresholds were used: 0.0001, 0.001, and 0.01.

Each fold-specific metabolite prediction model was tested on the corresponding testing fold to determine the correlation and R^2 between the predicted and actual metabolite values (Supplementary Figure 27). The mean predictive correlation was taken across all folds for each model, with the highest-correlated model chosen as the best predictive model for that metabolite. For each metabolite, the type of model, mean number of SNPs used, and presence of significant meta-analysis GWAS loci were recorded (Supplementary Table 7).” (lines 524-545)

11. Line 83, you say independent SNP as predictors. What is your definition/threshold to determine independency? Is it independent statistically? How did you select SNPs for prediction of each metabolite? Did you use their effect size from line 80?

The details on how independent SNPs were defined are laid out in the Methods section:

“The resulting fold-specific GWAS files were then clumped down to independent SNPs ($r^2 < 0.1$ within a 1000 kb window using the 1000 Genomes CEU reference panel for LD estimation) with a p-value threshold of 0.01 using PLINK.” (lines 521-523)

To clarify the general independence procedure used, the following text was added to the Methods Overview section:

“Genome-wide prediction models were built for each CSF metabolite with independent SNPs as predictors (based on SNP clumping), using both models...” (lines 89-90)

The selection of SNPs to predict each metabolite was based on which prediction method was used. The way PRS and ridge regression models work is by using all input SNPs as predictors. LASSO and elastic net models, on the other hand, induce sparsity in the set of predictors, excluding predictors that are not useful to the model. PRS models use the beta effect sizes from the GWAS as the SNP coefficients while the other models, being regression models, estimate SNP effect sizes when the model is fit.

References

1. Barbeira, A. N. *et al.* Exploring the phenotypic consequences of tissue specific gene expression variation inferred from GWAS summary statistics. *Nature Communications* **9**, 1825 (2018).
2. Bowden, J. *et al.* A framework for the investigation of pleiotropy in two-sample summary data Mendelian randomization. *Statistics in Medicine* **36**, 1783–1802 (2017).
3. Bowden, J., Davey Smith, G. & Burgess, S. Mendelian randomization with invalid instruments: effect estimation and bias detection through Egger regression. *Int J Epidemiol* **44**, 512–525 (2015).
4. Yu, B. *et al.* Genetic Determinants Influencing Human Serum Metabolome among African Americans. *PLOS Genet* **10**, e1004212 (2014).
5. Robinson, J. L. *et al.* An atlas of human metabolism. *Sci. Signal.* **13**, (2020).
6. Pardiñas, A. F. *et al.* Common schizophrenia alleles are enriched in mutation-intolerant genes and in regions under strong background selection. *Nat Genet* **50**, 381–389 (2018).
7. Ripke, S. *et al.* Biological insights from 108 schizophrenia-associated genetic loci. *Nature* **511**, 421–427 (2014).
8. Demontis, D. *et al.* Discovery of the first genome-wide significant risk loci for attention-deficit/hyperactivity disorder. *Nat Genet* **51**, 63–75 (2019).
9. Lee, J. J. *et al.* Gene discovery and polygenic prediction from a 1.1-million-person GWAS of educational attainment. *Nat Genet* **50**, 1112–1121 (2018).
10. Hemani, G. *et al.* The MR-Base platform supports systematic causal inference across the human phenome. *eLife* **7**, e34408 (2018).

11. Jones, S. E. *et al.* Genome-Wide Association Analyses in 128,266 Individuals Identifies New Morningness and Sleep Duration Loci. *PLOS Genet* **12**, e1006125 (2016).
12. Elsworth, B. L. *et al.* *MRC IEU UK Biobank GWAS pipeline version 2.* (2019).
13. Kang, H., Zhang, A., Cai, T. T. & Small, D. S. Instrumental Variables Estimation With Some Invalid Instruments and its Application to Mendelian Randomization. *Journal of the American Statistical Association* **111**, 132–144 (2016).
14. Bowden, J., Davey Smith, G., Haycock, P. C. & Burgess, S. Consistent Estimation in Mendelian Randomization with Some Invalid Instruments Using a Weighted Median Estimator. *Genet Epidemiol* **40**, 304–314 (2016).
15. Willer, C. J., Li, Y. & Abecasis, G. R. METAL: fast and efficient meta-analysis of genomewide association scans. *Bioinformatics* **26**, 2190–2191 (2010).
16. Gamazon, E. R. *et al.* A gene-based association method for mapping traits using reference transcriptome data. *Nat Genet* **47**, 1091–1098 (2015).
17. Tibshirani, R. Regression Shrinkage and Selection via the Lasso. *J Roy Stat Soc B Met* **58**, 267–288 (1996).
18. Zou, H. & Hastie, T. Regularization and variable selection via the elastic net. *J Roy Stat Soc B* **67**, 301–320 (2005).
19. Hoerl, A. E. & Kennard, R. W. Ridge Regression: Biased Estimation for Nonorthogonal Problems. *Technometrics* **12**, 55–67 (1970).
20. Dudbridge, F. Power and Predictive Accuracy of Polygenic Risk Scores. *PLOS Genet* **9**, e1003348 (2013).
21. Friedman, J., Hastie, T. & Tibshirani, R. Regularization Paths for Generalized Linear Models via Coordinate Descent. *J Stat Softw* **33**, 1–22 (2010).

22. Euesden, J., Lewis, C. M. & O'Reilly, P. F. PRSice: Polygenic Risk Score software. *Bioinformatics* **31**, 1466–1468 (2015).

REVIEWERS' COMMENTS:

Reviewer #2 (Remarks to the Author):

A flowchart for lines 574-588 is required. I asked it in my previous comments. If you have provided it, please direct me there. The supplementary Figure 1 is not sufficient. At least, in the method section a flowchart that explains the steps for two-sample MR is necessary. The explanations in the body is not informative at it is. I have a very hard time to follow the steps. You may want to use my suggestion below:

Since multiple data sets are used, you may want to put in the flowchart as sample 1(SNP and metabolite), sample 2 (SNP and phenotype). Then, show SNPs from which sample is used as an IV in the other sample; and etc.

Response to reviewer comments

Note: added text is in red.

Response to Reviewer #2

- 1. A flowchart for lines 574-588 is required. I asked it in my previous comments. If you have provided it, please direct me there. The supplementary Figure 1 is not sufficient. At least, in the method section a flowchart that explains the steps for two-sample MR is necessary. The explanations in the body is not informative at it is. I have a very hard time to follow the steps. You may want to use my suggestion below:**

Since multiple data sets are used, you may want to put in the flowchart as sample 1 (SNP and metabolite), sample 2 (SNP and phenotype). Then, show SNPs from which sample is used as an IV in the other sample; and etc.

We have added a new supplementary figure, Supplementary Figure 29 (copied below, referenced in the appropriate Methods section) that highlights where the data sets come from for each part of the 2-sample MR. In the caption for this figure, we additionally explain how SNPs were chosen for the MR analyses.

Supplementary Figure 29. Two-sample Mendelian Randomization model set-up

An overview of the sources of each data set used in the two-sample Mendelian Randomization analysis. The set of metabolite-phenotype pairs to analyze was the set of significant metabolite-phenotype associations from the MWAS analysis. For each subsequent MR analysis, the set of instrument SNPs for each metabolite were those that were in the best predictive model for that metabolite from the MWAS analysis; the GWAS summary statistics for the metabolites were those from the WADRC/WRAP GWAS meta-analysis; and the GWAS summary statistics for the phenotypes were taken from summary statistics in the IEU GWAS Database (Supplementary Table 12).